

# An insight into pre-Columbian raised fields: The case of San Borja, Bolivian lowlands

Leonor Rodrigues [1], Umberto Lombardo[2], Mareike Trauerstein[1], Perrine Huber[3], Sandra Mohr[3], Heinz Veit[1]

[1] Institute of Geography, University of Berne, Hallerstrasse 12, CH-3012 Bern.
[2] University of Pompeu Fabra, Ramon Trias Fargas 25-27, Mercè Rodoreda ES-08005 Barcelona.
[3] Independent researcher, CH-3012 Bern.

*Correspondence to*: Leonor Rodrigues (Leonor.rodrigues@giub.unibe.ch)

**Abstract.** Pre-Columbian raised field agriculture in the tropical lowlands of South America has received increasing attention and been the focus of heated debates regarding its function, productivity, and role in the development of pre-Columbian

societies. Even though raised fields are all associated to permanent or semi-permanent high water levels, they occur in different environmental contexts. Very few field-based studies on raised fields have been carried out in the tropical lowlands and little is known about their use and past management. Based on topographic surveying and mapping, soil physical/chemical analysis and OSL and radiocarbon dating, this paper provides insight into the morphology, functioning and time frame of the use of raised fields in the south-western Llanos de Moxos, Bolivian Amazon. We have studied raised fields of different sizes that

were built in an area, near the town of San Borja, with a complex fluvial history. The results show that differences in field size and height are the result of an adaptation to a site where soil properties vary significantly on a scale of tens to hundreds of meters. The analysis and dating of the raised fields sediments point towards an extensive and rather brief use of the raised fields, for about 100-200 years at the beginning of the 2nd millennium.

## 1 Introduction

The Llanos de Moxos (LM) is one of the largest floodplains in Latin America, characterized by a prolonged rainy season and a contrasting dry season, often resulting in either severe flooding or droughts (Hanagarth, 1993). Nowadays, the LM is sparsely inhabited, soils have been considered unsuitable for agriculture and the main economic activity is extensive cattle grazing (Erickson, 2003b). However, the presence of numerous pre-Columbian earthworks indicates that in the past, humans modified the landscape in several ways and cultivated crops (Denevan, 2001; Erickson, 2008; Lombardo et al., 2011b; Jaimes

Betancourt, 2013; Prümers and Jaimes Betancourt, 2014; Carson et al., 2015). The number and variety of earthworks has made the area of the LM one of the most important examples of pre-Columbian anthropogenic landscapes in the Amazon Basin (Erickson, 2008). Raised fields are one of the most abundant and impressive types of earthworks found in the LM. Pre-Columbian raised fields are elevated agricultural earth platforms. They are also found in a number of other Latin American countries. Distributed over a wide range of latitudes, spanning from almost 20°N to 40°S, they exist in very different

environmental contexts: in seasonal inundated floodplains, around lake shores and in coastal zones; all of them associated to



permanent or semi-permanent high water levels (Erickson, 1988; Kolata and Ortloff, 1989; Wilson et al., 2002; Dillehay et al., 2007; Denevan, 2001; Rostain, 2010). Since their discovery about 100 years ago by Erland Nordenskiöld (Denevan, 2009), increasing interest in raised field agriculture has led to a body of literature focused on understanding how raised fields worked and were managed and how pre-Columbian societies were sustained (Gliessman, 1983; Kolata and Ortloff, 1989; Erickson,

2008; Iriarte et al., 2010; McKey et al., 2010; Rostain, 2010; Lombardo et al., 2011a; Renard et al., 2011; Whitney et al., 2014; Rodrigues et al., 2015; Walker, 2004). Nevertheless, there is still limited data and a lack of consensus about the time frame during which raised fields were in use, why they were built and how they were managed. Some studies have suggested that raised fields were highly productive and could have sustained dense populations in the LM (Erickson, 2008), and the technology of raised fields has been proposed as a model for sustainable agriculture today (Erickson, 2008; Renard et al., 2012;

Whitney et al., 2014). In contrast, other studies suggest that raised fields in the LM were not more productive *per se* and were rather constructed to overcome past climate extremes (Lombardo et al., 2011a; Rodrigues et al., 2016).

Experimental studies have aimed to assess the productivity of raised fields and, because of their alleged high yields, they have been presented as a "sustainable agriculture" alternative for rural development projects (Erickson, 1992; Stab and Arce, 2000; Saavedra, 2009). However, most of these rehabilitation projects failed due to a number of reasons: overestimation of field

productivity (Bandy, 2005), unfavourable structure of modern society (Erickson, 2003a), environmental conditions (Chapin, 1988), non-involvement of local communities due to a top-down approach (Renard et al., 2012), other methodological weaknesses (Lombardo et al., 2011a) and the generalization of one single technology as a model for all raised fields regardless of their location (Baveye, 2013). As already stated, raised fields are highly diverse in design and exist in very different environmental contexts. Therefore, it has been proposed that the way they were built and their management were probably

determined by geographical/environmental constraints (Denevan, 2001; McKey et al., 2010; Lombardo et al., 2011a; Baveye, 2013; Rostain, 2013; Rodrigues et al., 2016). However, the reasons behind the regional differences in raised field types is still not clear, as differences in raised fields could also be related to cultural diversity (Erickson, 1995; Denevan, 2001; Walker, 2011; Rostain, 2013).

Understanding the link between environmental variables (e.g. soils, topography, and hydrology) and design of raised fields is

key in order to better understand the reasons behind raised field diversity and to infer how they could have been managed in the past. Very little is known with regards to the environmental characteristics of the areas where raised fields were built in lowland South America. Only recently some studies have started to include detailed soil studies providing an insight to the internal structure of raised fields and their relation to the environment (Iriarte et al., 2010; McKey et al., 2010; Lombardo et al., 2011a; McMichael et al., 2014; Rodrigues et al., 2015).

The LM is very suitable for the study of raised fields, as they appear in a variety of forms (ridged fields, platform fields, mound fields, ditched fields) and patterns (e.g. parallel or randomly scattered, with or without embankment and causeways). These different types of fields tend to be present in different parts of the LM and do not normally coexist in the same site (Denevan, 2001; Lombardo et al., 2011b). Nevertheless, recently it has been shown that, in at least two cases in the LM, differences in





shape, height and layout can vary considerably within the same cultural area, and are the result of an adaptation to the distinct local edaphology (Lombardo et al., 2011a; Rodrigues et al., 2016).

The present study explores the raised fields located in the savannah on the western bank of the Rio Maniqui, in the San Borja area (Fig. 1). Archaeological evidence of occupational sites and raised fields here were already described by Nordenskiöld (1916) at the beginning of the 20[th] century and again recently by Iriarte and Dickau (2012). Up till now no data from archaeological excavations have been available for this area. Here, raised fields of different shapes co-exist. This study aims to further our understanding of pre-Columbian agricultural systems in the Bolivian lowlands. It uses topographic surveying and mapping, soil physical/chemical analysis and OSL and radiocarbon dating to address the following questions:

- In which environmental context can raised fields be found and what are their characteristics (e.g. morphology and soil properties)?
- Are there links between the dimension/shape of raised fields and soil properties?
- When were these raised fields in use?

## 2 Study area

### 2.1 Geography and environment

The study area is located near San Borja, a town situated in the southwestern part of the Beni department, only a few kilometres away from the Andean foothills (Fig. 1). The Beni department almost completely overlaps with the Llanos de Moxos (LM), a seasonally inundated floodplain drained by three major rivers: Río Mamoré, Río Beni and Río Iténez. The diverse geomorphology of the LM is shaped by past and present fluvial dynamics such as alluvial deposition and erosion and river shifting (Hanagarth, 1993; Dumont and Fournier, 1994; May, 2011; Plotzki et al., 2011; Plotzki et al., 2013; Lombardo, 2015), as well as tectonics (Hanagarth, 1993; Dumont and Fournier, 1994; Lombardo, 2014). These processes are responsible for changes in the local topography, determining the flooding dynamics and, in turn, the forest-savannah ecotone (Mayle et al., 2007). The climate in the LM is controlled by the South American Summer Monsoon, leading to heavy convective rainfalls in austral summer and dry conditions in winter (Zhou and Lau, 1998; Garreaud et al., 2009). The mean annual temperature is 25.8° C and is fairly stable year round (Navarro and Maldonado, 2002), although it can decrease considerably during the austral winter months with the arrival of cold southern winds locally called *surazos* (Espinoza et al., 2013). The mean annual precipitation in the region amounts to 1900 mm (Hijmanns et al., 2005); most of the precipitation falls during the austral summer, followed by prolonged flooding which covers an area ranging from 30,000 km$^2$ to >80,000 km$^2$ (Hanagarth, 1993; Hamilton et al., 2004). In general floods are greater in the southern LM due to increased precipitation towards the Andes and higher groundwater tables (Hanagarth, 1993). The landscape around San Borja has been shaped by the fluvial history of the Río Maniqui, a meandering white water river which has shifted its course several times in the past. Hanagarth (2003) has distinguished nine phases of ancient river courses; the Maniqui has been shifting its course in both clockwise and anti-



clockwise directions. The Río Maniqui is one of the most dynamic rivers in the LM (Lombardo, 2015) with crevasses occurring every few years, leading to complete avulsions on a sub decadal time frame. Throughout the Holocene, this river, with a high sedimentary load, has built a large interior delta (Hanagarth, 1993; Lombardo, 2014). An extensive forest has grown on this delta, which today forms part of the Biosphere "Estación Biológica del Beni" (Hanagarth, 1993).

Soils along the Río Maniqui are in general acidic, with a pH ranging from 3.86 to 5.11, but they are very heterogeneous in terms of plant available nutrients, mostly correlated to the particle size of the sediments (Guèze et al, 2013). Soils in the southern LM are generally loam or silty loam and silty clay loam (Boixadera et al., 2003), all soils are subject to hydromorphic processes and are mainly acidic (Boixadera et al., 2003; Rodrigues et al., 2015), with some exceptions of saline soils (pH of >8) (Lombardo et al., 2015) and soils with accumulation of calcium carbonates in the subsoil (Boixadera et al., 2003).

**2.2 The archaeology of raised fields in the Llanos de Moxos**

Our knowledge about the chronology, complexity and evolution of pre-Columbian cultures in the LM is still in its early stages (Prümers and Jaimes Betancourt, 2014). The archaeological landscape is roughly divided by the Río Mamoré; on the western side no raised fields exist, while on the eastern side thousands of hectares of fields have been documented (Denevan, 2001; Lombardo et al., 2011b). Detailed archaeological research has mostly concentrated in the Monumental Mounds Region (MMR)

(Prümers, 2008; Bruno, 2010; Dickau et al., 2012; Jaimes Betancourt, 2012; Lombardo and Prümers, 2010; Lombardo et al., 2013), in the southwestern LM. Here, no raised fields are present. Unfortunately, almost no chronological data exists for the eastern Llanos, where fields are widespread. Up till now, habitational sites associated to raised fields have only been dated in four locations: the San Juan site (AD 446-613) and the Cerro site (AD 1300-1400), in the northern part of the LM (Walker, 2004), and the Moxitania site (AD 700-1000), Abularach and Carretera Santa Ana sites (AD 900-1100), close to San Ignacio

de Moxos (Villalba et al., 2004). Raised fields in Bermeo, close to San Ignacio, have been dated; they were used intermittently from AD 570–770 up to the 14th century (Rodrigues et al., 2015).

**3 Methods**

**3.1 Mapping and field work**

The location for the study of raised fields was first predefined with the help of Google Earth. An area of about 8 ha was

selected, where raised fields of different shape coexist (Fig.1) .ArcGIS was used to map natural- and anthropogenic features at two different scales (Google Earth 2002, 2011). On a large scale, covering an area of 2500 $km^2$, paleo-channels and areas with raised fields and causeways have been mapped (Fig. 2a), including topographic information using CGIAR-CSI SRTM (resolution 90 m) (Fig. 2b). On a smaller scale (12.3 $km^2$), in the area where field work was conducted, individual raised fields, causeways, ponds, settlement sites, as well as natural features like palaeo-channels and creeks, were mapped (Fig. 3).

Field work was conducted in August 2012 and again in 2013. The local relief was measured using a digital level Sokkia D50. A 480 m long topographic transect perpendicular to the fields was drawn based on measurements taken approximately every




1 m (Fig. 7). In addition, a specific area where higher and lower fields lie next to each other was selected for in depth morphological analysis: 2400 elevation points were measured and a digital elevation model (DEM) was generated using the 3D analyst extension of ArcGis with natural neighbour interpolation (Fig. 7). Four fields were excavated; trenches were dug from the ridge to the canal (Fields 1-4). Two additional pits were dug; one in an area away from the raised fields (No Field, NF) and one in a causeway (CW) (Fig. 1). In total 10 stratigraphic profiles were prepared and sampled every 10 cm: the NF profile, the CW profile and two for each Field (1-4), one profile in the ridge and a second one in the adjoining canal. The description of the horizons/layers follow the guidelines of FAO (IUSS Working Group WRB, 2014). In addition, a virtual grid was applied onto an area of 450x64 m and samples for particle size analysis were taken every 16 m in the N-S direction and every 20 m in the E-W direction (Fig. 1 and 6). Particle size was analysed every 20 cm up to a depth of 100 cm.

## 3.2 Laboratory analysis

All the samples have been air-dried. The colour determination, however, was carried out on moist samples, using the Munsell soil color charts (1994). For particle size distribution, organic matter was removed with 30% $H_2O_2$ and afterwards measured with a laser diffraction instrument (Malvern Mastersizer Hydro 2000S). The pH was measured with a glass-electrode after mixing the sample with a 0.01 M $CaCl_2$ solution. C and N concentrations were analysed by dry combustion and gas chromatographic separation with a CNS analyser (vario El cube). Cation Exchange Capacity (CEC) was measured by means of extraction of the exchangeable Cations $Ca^{2+}$, $Mg^{2+}$, $K^+$, $Na^+$, $Mn^{2+}$, and $Al^{3+}$, with 1M ammonium nitrate solution ($NH_4NO_3$). Concentrations were measured using an atomic absorption spectrometer of the type analytikjena ZEEnit 700P. Effective Cation Exchange Capacity ($CEC_{eff}$) and Base Saturation (Bs) were calculated as follows: $CEC_{eff} = \sum$ Cations (exchangeable Cation = Cation mg/kg/ molar mass mmol/l * valence) and $BS = ([Ca] + [Mg] + [K] + [Na]/ CEC_{eff})*100$. Available phosphorus ($P_{av}$) was measured with the Mehlich I extraction method (Mehlich, 1953), following the guidelines recommended for acidic soils in Sparks (1996) and standard analysis for tropical soils in Brazil (Solos, 2011). $P_{av}$ content was determined by means of a spectrophotometer reacting with ammonium molybdate, following the slightly modified method of Murphy and Riley (1962) developed by Watanabe and Olsen, SR (1965). Element analysis has been performed by x-ray fluorescence spectroscopy (XRF) and quantified by means of the *UniQuant* method.

## 3.3 Radiocarbon- and Optically Stimulated Luminescence (OSL) dating

AMS C analysis on three charcoal samples and two palaeo-soil samples (Tab.1) was conducted at the Poznan Radiocarbon Laboratory and LARA AMS Laboratory in Bern, calibrated using Calib 7.1 (Stuiver and Reimer, 1993) and the SHcal13 calibration curve for the southern hemisphere (Hogg et al., 2013).

Samples for Optically Stimulated Luminescence (OSL) dating were taken by pushing steel tubes into the exposed sediment. The concentration of dose rate relevant elements (Table S1) was determined using high- resolution low-level gamma spectrometry, performed on bulk material from the surrounding sediment. Dose rates have been calculated assuming an average moisture content of 25-35 % and present day sediment cover. For equivalent dose ($D_e$) determination, samples were dry sieved



to separate the 100–150 μm particle size fraction, followed by HCl and H2O2 treatment. The quartz fraction was extracted using heavy liquids and etching in 40% HF for 1 h. Luminescence measurements were carried out on a Risø DA-20 TL/OSL reader fitted with an internal 90Sr/90Y beta-source. Quartz signals were detected through a Hoya U340 detection filter. De measurements were performed on 48 small aliquots (2mm) per sample applying the Single Aliquot Regenerative-dose (SAR)

protocol (Murray and Wintle, 2000) using a preheat of 230 °C for 10 s prior to all OSL measurements. The small aliquot OSL signals show no indication for feldspar contamination (IR depletion ratio > 0.8) and are dominated by the fast component. Small aliquot $D_e$ distributions of sample SB1-C36 and SB1-C40 are slightly skewed, with some outliers at the upper end of the distribution, indicating partial bleaching. To exclude signal averaging, single grain measurements were additionally carried out on all samples using the same measurements parameters as for the small aliquots. For the 2 sediment samples (SB1 C60

and SB1 F140), single grain $D_e$ distributions are symmetric and exhibit an overdispersion of 28 and 21%, respectively. The resulting single grain CAM ages are consistent with the small aliquot ones (see Table S1). The single grain $D_e$ distributions for the samples from the field/canal deposits (SB1-C36 and SB1-C40) are skewed and exhibit an overdispersion of 66 and 62%, respectively. To calculate MAM ages (Galbraith et al., 1999) from the single grain data of these samples a *sigma b* value of 0.28 was applied. In the following discussion single grain ages of all samples are used.

**4 Results**

**4.1 Field work and mapping of the study site:**

The area studied has been shaped by several shifting rivers, and more recently by anthropogenic earthworks. Natural features like palaeo-channels and oxbows, and anthropogenic earthworks, including raised fields and causeways, are illustrated in Figure 2. Several generations of palaeo-river channels were clearly identified; these share the direction of the modern Río

Maniqui, from the southeast to the northwest. One paleo-river can be traced continuously while in the other cases only segments of paleo-channels can be recognized. Channels and oxbows often appear washed out due to enduring erosion, the superimposing of other channels and the construction of earthworks, mostly raised fields. Patches of gallery forest exist along the channels and are recognisable by the topographic differences in Fig. 2b. In total, 370 ha of raised fields were mapped, as well as 52.2 km of causeways (Fig. 2a). All the raised fields are found along palaeo-rivers, on alluvial deposits, and are

associated to causeways, but not vice versa. Some causeways were built through the pampa, connecting higher laying areas covered by forest along palaeo- rivers (Fig. 2).

On a smaller area, detailed structures like small creeks, point bars, anthropogenic features, including raised fields, causeways, settlement sites and ponds, were mapped (Fig. 3). The natural features can be categorized as continuous meander channels, creeks, oxbows and point bars. The assignment of the oxbows to the larger channels is not straight forward as they sometimes

display similar channel width and sometimes they differ by several meters. It is therefore difficult to say if oxbows and larger channels have been formed by the same river. In some cases oxbows are dried out, but most of them exist as wetlands. Point bars are found next to the larger channels and oxbows. During the field survey we mapped four little earth mounds, locally





called *lomas,* and one pond (Fig. 3). Lomas are anthropogenic earth mounds found in the lowlands of Bolivia, which could have served several purposes: as settlements, cemeteries, ritual sites and, as well, for agriculture (Erickson, 2000). The *lomas* surveyed are located on the most elevated parts of the study site, near two farms: *Campo España* and *El Progresso* (Fig. 3) Within this area individual raised fields were mapped, as well as 14577 m of causeways (Fig. 3). The longest causeway,

crossing the whole area from the southwest to the northeast, is 2997 m long. From this major causeway several shorter causeways go north and south, always in connection with raised fields. The shape of the raised fields, as well as the causeways, are particularly interesting. Most of the fields (>100 m) and causeways share a curved course, which is similar to the shape of the meanders and point bars of the palaeo-rivers. Some fields (< 100 m) were built perpendicular to a causeway, resembling a comb, a pattern already described for raised fields in the Titicaca Basin (Denevan, 2001). They are all elongated, with a mean

length of 87 m; however, their length varies considerably, ranging from a minimum of 9 m to a maximum of 582 m.

## 4.2 Sedimentary and pedogenic characteristics

### 4.2.1 Individual Profiles

As field work was conducted twice, in 2012 and 2013, and the rainy season during 2013 was much wetter, the depth of the groundwater table was significantly different each year; at the beginning of August 2012 the water table was 2 m below the

surface, whereas at the end of July 2013 it was at a depth of 1 m. Detailed profile descriptions are summarized in Figure 4. All profiles share some pedogenic characteristics, but in general they differ remarkably in a number of aspects. All profiles show intense mottling, typical of hydromorphic processes. The iron mottles are soft, up to 1.5 cm in diameter and the colour varies between yellow and orange, which usually indicates the presence of Goethite and Lepidocrocite, typically formed in waterlogged soils (Cornell and Schwertmann, 2003). Black mm-scale manganese concretions were also observed.

Hydromorphy is present, on average, up to 35 cm below the top of the ridge, but there is no clear boundary. Manganese concretions tend to accumulate at the upper limits of the hydromorphic affected layers. The depth of this diffuse boundary slightly differs from field to field and is indicated with a blue line in Figure 4. With regards to the canals, the profiles can generally be divided into two major layers; the infilling of the canal after the construction of the fields and the undisturbed sediments below. This sharp boundary can be clearly recognized due to the brown infilled canals contrasting with the yellowish

sediments below. This boundary is always present around 30-50 cm depth (Fig. 4). With regards to the texture, all profiles differ remarkably (Fig. 5 and Tab. 1) and distinctive characteristics were observed for each field.

**Field 1.** Soil texture in Field 1 is mainly loam-silty loam. Particle size decreases towards the bottom, ranging from 52% to only 27% of sand. During the excavation two small pieces of burnt earth were found.

**Fields 2 and 3.** The soil in these fields is relatively coarse in texture, ranging from sand to sandy loam, with 65-90% sand and

only 1.5-8 % clay. Particle size increases towards the bottom of the fields, from sand to sandy loam. Both profiles comprise a ferruginous yellow-orange continuous line (Fig. 4). This line originates at the boundary of the infilling of the canal, following the topography of the field about 40 cm below the surface, and disappears towards the top/middle of the ridge. One charcoal





piece (diameter 5 mm) was extracted from the ridge of Field 2 at a depth of 80 cm. Beneath the canal of Field 2 a palaeosol (Palaeosol I), recognisable by its dark brown colour, was found at a depth of 270 cm using a hand auger (Tab. 2).

**Field 4.** The soil here has the finest texture of all raised fields, ranging from silt to silty loam. There is a significant increase in clay content from 10% to 26% towards the bottom, whilst sand decreases from 41% to 6%. A lot of natural and anthropogenic disturbance is evident. In the upper layers some clear signs of modern deformation by cattle trampling can be recognized (Fig 4). Further down in the ridge profile desiccation cracks are filled with fine sand. The relative high amount of clay in the deeper layers (80-100 cm) are responsible for the cracks, which commonly develop by the expansion and shrinking of the clays due to repeated wetting and drying (Schachtschabel et al., 2002).

**No Field Profile (NF). The soil of the profile from the area without raised fields** is extremely dense and has the finest texture, with up to 32% clay, ranging from silt loam to silty clay loam. A palaeosol (Palaeosol II) was detected at a depth of 60-70 cm, recognisable by its darker brownish colour (Fig. 4).

**Causeway (CW). The profile from the causeway** differs in many aspects from the others profiles. The CW in general is much denser and was hard to excavate. The texture is loam (27-52% sand and 8-15% clay). Hydromorphic features are almost absent up to a depth of 50 cm, where density increases remarkably and there is a sharp colour change from light yellow-brown to darker brown. This change may be interpreted as the ancient surface upon which sediments were heaped up to construct the causeway. The bottom part of the profile, starting at around 50 cm, is affected by hydromorphism, characterized by few manganese concretions and about 20% of iron mottling. There are a striking amount of pieces of charcoal at a depth of 30 -50 cm, which were almost completely absent in the other profiles.

### 4.2.2 Particle size distribution from grid sampling

Results from the grid sampling shows that most of the sediments contain a high percentage of silt, the content of sand with respect to clay is very heterogeneous. The proportion of sand varies considerably from 2% to 41% (compare with Fig. 5). The median particle size of each sample (d50 µm) has been used to illustrate particle size distribution at each depth, as it has shown to best reflect the differences within the area (Fig. 6).

One south-north oriented sand structure which is partly buried by fine sediments is evident in column 13. Two additional sand structures, one in column 11 and the other in column 7, are visible at the border of the sampled area; they seem to have the same south-north orientation. The finest texture of silty clay loam can be found, at both west and east ends, in columns 0-3 and 18-20 respectively, where no fields were built (Fig. 6).

For all the fields, down-profile particle size distribution was measured separately and the results are consistent with the results from the grid. The fields with a finer texture of silt loam (Fields 1 and 4) show a coarsening up profile, where in the upper 20 cm there is considerably more sand compared to the bottom. In contrast, the profiles in the middle columns 10-13 show, on top of the sand structures, a fining upward sequence (Field 2 and 3). The highest fields are all built on the coarsest textures (row A, columns 8-15), while the lower smaller fields were mainly built on silty loam (row B-D, columns 3-18). It is striking



that in the upper 60 cm of Field 3 the texture changes along its course towards the southeast, starting with sandy loam at the point of the excavation of the raised fields and ending up with silty loam towards the causeway.

Groups of parallel fields with similar heights were identified; these are always separated by a causeway (Fig. 7). The fields we excavated (Fields 1-4 in Fig. 7) were built on a slight slope. A topographic transect going from west to east reveals a downward trend, with a maximum difference of 90 cm (Fig. 6 and 7).

The highest points were measured on the ridge of Field 2 and the causeways (Fig. 6b). The difference between ridge and canal for most of the fields is around 25 cm, but in some fields it is up to 60 cm (Field 2). The DEM, which includes the higher and the lower fields, illustrates the difference in height and shows that the larger fields are, on average, 50% taller than the smaller fields (Fig. 6c). The higher fields are much better preserved than the lower fields, because the latter have been partially destroyed by cattle trampling. There are several causeways; one major causeway going from the southwest to northeast, three perpendicular to it and one cutting all the others, going from the southeast to northwest. The latter separates the lower fields from the higher ones (Fig. 6c). The major causeway connects the area of the raised fields with a settlement area and the *lomas,* while the other causeways are only linked to raised fields (compare with Fig.3).

### 4.3 Geochemistry

The sediments elemental composition is poor in carbonates, with less than 0.1 % of CaO. The sediments are dominated by Quartz ($SiO_2$), with a mean percentage of 76% for Field 1, 85% for Field 3, 70% for Field 4 and 63% for NF, followed by a mean percentage of aluminium ($Al_2O_3$) of 12%, 7%, 14% and 17% respectively. For all the ridge profiles the more soluble elements (Na, $K_2O$, CaO and MgO as well as $Fe_2O_3$ and $Al_2O_3$ oxides) increase towards the bottom (Fig. 8). Some anomalies can be observed where Mn, $P_2O_5$ and $Fe_2O_3$ are more concentrated in specific layers. A clear enrichment of Mn always occurs at the top of all profiles, except for Canal 4 where $Fe_2O_3$ is accumulated instead. It is surprising that $P_2O_5$, which is always concentrated in the relatively upper part (20-45 cm) of the canals, in the fields the highest concentrations are found deeper in the profile (40-70 cm). This enrichment of $PO_4$ is clearly accompanied by higher Fe/Al and Fe/Ti, indicating a higher level of ferruginisation (relative accumulation of Iron) (McQueen, 2006) (Fig. 7 and Table S3). In NF there is no important accumulation of $P_2O_5$, however, it does increase towards the bottom. $P_2O_5$ in the NF Profile is slightly lower, very high values of CaO and Na have been detected in Palaeosol II, at a depth of 60-70 cm (Fig. 6).

Further soil chemical properties ($C_{org}$, N, $CEC_{eff}$, pH, Bs, $P_{av}$) describing soil fertility are summarized in Tab. 1 and partly illustrated in Fig. 9. In all profiles $C_{org}$ decreases with depth. Values in the ridges range from 0.74% to 0.01% towards the bottom and in the canals from 1.13% to 0.01%. In the ridges $C_{org}$ decreases smoothly, whereas in the canals an abrupt change occurs always at the boundary of the infilling. An exception is profile NF, where at a depth of 60-70 cm there is an increase of $C_{org}$ that reaches 0.34% due to the presence of the palaeosol (Palaeosol II). In general $C_{org}$ values for the profile NF are high, going from 1.31% to 0.21 %. In all profiles the C/N values in the first centimetres range from 7.7 to 10.57, which are typical values for the tropics, where organic matter is mineralized fast (Schachtschabel et al., 2002). The exchangeable cation ($CEC_{eff}$)





and pH for each profile are given in Table 1. Considerable differences are evident: according to the criteria of Hazelton and Murphy (2007), $CEC_{eff}$ is very low in Fields 2 and 3, low in Field 1 and low to moderate in Field 4 and NF. Base saturation is very low for Fields 2 and 3, whereas levels are more favourable in 1, 4 and NF, with moderate (40-60%) to very high (80-100%). Base saturation (Tab. 1 and Fig. 9).

Available phosphorous ($P_{av}$) is generally low, but amounts are highly variable going from below limit of detection to 23 ppm (Tab. 1 and Fig. 9). According to the criteria of Hazelton and Murphy (2007), $P_{av}$ values in the upper parts of the ridges can be classified as low (<5 ppm), but become moderate (10-17ppm) towards the bottom. There are some very high values in Field 1, at the bottom of the ridge and the canal (20 ppm / 26 ppm) and in the canal belonging to Field 3 (21 ppm) at a depth of 55-

10 65 cm. Here, the available phosphorous ($P_{av}$) accounts for up to1% of the total phosphorous, whereas in the other samples $P_{av}$ is <0.5%. To assess the relationship of the parameters describing the fertility of the sediments a pearson correlation has been performed for all samples (Fig.10)

$CEC_{eff}$ and Bs values correlate with particle size, whereas $P_{av}$ does not (Fig. 10). In general the $P_{av}$ values are strongly associated

to the total amount of $P_2O_5$ which is accumulated in a specific layer (compare Tab. 1 with Fig.8). The pH is acidic in all profiles and is correlated to the total amount of Ca. This can be clearly seen in Field 4 and NF, where the pH increases together with CaO. The lowest PH values, which do not exceed 4.2, can be found in the sandiest Fields 2 and 3, with low Bs and a high percentage of exchangeable Al. In these fields the Ca/Al ratio is < 1.0 and aluminium toxicity can be a problem for plants (Cronan and Grigal, 1995). In contrast, in Field 4, which is composed of much finer sediments (silt-silt loam), the pH values

reach 5.2, Bs is high, with low to no exchangeable Al and the Ca/Al ratio is > 1.0.

### 4.4 Chronological framework

A total of four OSL and five radiocarbon ages have been obtained (Tab. 2, Fig.11). Two OSL ages were taken from the ridge profile 2 (40 cm and 140 cm) and two from the adjoining canal (36 cm and 60 cm). Samples for radiocarbon dating were taken

from Palaeosol I, at a depth of270 cm below Field 2 and from Palaeosol II, below NF, at a depth of 65 cm. The three remaining radiocarbon ages are from charcoal pieces: two extracted from the excavated CW and one from the ridge of Field 2. The oldest age, cal BC 5212- 4940, was found in the Palaeosol I, beneath the canal of Field 2 (270 cm depth). In this case only the humin fraction could be dated, as humates were missing. In contrast, both fractions could be dated in the case of Palaesol II in the NF profile (65 cm). The difference between the age of the humins (cal BC 5934-5775) and the much younger age of the humates

(cal BC 976-816) is significant and might point towards a contamination (Walker, 2005). The OSL ages are consistent with the stratigraphy, the two basal ages are BC 3500-2680 for the ridge and BC 2870-2130 for the canal and the top ages are AD 1150-1290 and AD 1320-1430, respectively. The radiocarbon ages of two charcoal pieces extracted from the CW were BC 797-751 and cal BC 743-687 and from Field 2 BC 1438-1262.



## 5 Discussion

All the fields mapped in the vicinity of San Borja show two essential characteristics which could help explain why they were built and their shape and size.

Firstly, all the fields were constructed on fluvial deposits, which are naturally higher than the pampa, and are made of relatively
coarse, hence better drained, sediments. Secondly, while field height seems to be related to sediment characteristics, distribution pattern seems to depend on the natural landscapes morphology.

### 5.1 Main characteristics of raised fields versus the landscape

Existing studies of raised fields in the LM have similarly shown that the majority of raised fields were built mainly on fluvial levees and on the naturally well drained areas, often on silty to sandy sediments, with the aim of improving drainage (Walker,
2004; Lombardo et al., 2011a; Rodrigues et al., 2015). It seems that pre-Columbian people took advantage of the natural morphology of the rivers and built raised fields on the levees or point bars, where the coarser sandy sediments are deposited. Areas where surface sediments were too fine were avoided (Fig. 6). It should be noted that no connection exists between the height of the raised fields and the general topography, in fact, the lowest fields (for example Field 4) are located in the lowest lying areas. The soil properties of the raised fields studied differ considerably, probably due to the heterogeneity of the
sediments in the area. Frequent changes in soil texture may be explained by the high frequency of crevasses and avulsions of the Maniqui River (Lombardo 2015). The analysis of satellite image and particle size distribution shows that the landscape history of the area is complex, resulting from a combination of several palaeo-river generations of different dimensions and the depositional behaviour of the meandering rivers. On such diverse landscapes sediment properties can therefore vary naturally within the range of meters, in this case resulting in the great variability of sediments comprising the excavated fields.

### 5.2 Age and morphology of raised fields

#### 5.2.1 OSL

OSL dating of Field 2 shows three major time phases (Fig. 11): Phase 1 comprises the oldest ages, corresponding to the deposition of the fluvial sediments, phase 2 indicates the period during which the field was built/used and phase 3 marks the time when the raised field was abandoned.
The top layers include an age of AD 1150-1290 for the ridge of Field 2 and AD 1320-1430 for the adjoining Canal (Fig. 10, Tab. 2). The OSL age refers to the moment in which quartz grains were buried, therefore the age AD 1150-1290 probably indicates the moment in which Field 2 was built. In Canal 2, the age AD 1320-1430 marks the time of the canal infilling. This implies that there was no further digging of the canal afterwards, suggesting the abandonment of the fields. The estimated time of abandonment, AD 1320-1430, is consistent with the abandonment of the raised fields studied in Bermeo, around AD 1400
(Rodrigues et al., 2015). The exact time of construction and abandonment of the fields, however, cannot be deduced from these ages, as the fields could have been elevated more than once (Rodrigues et al., 2015). Furthermore, the arrangement of fields





in groups separated by causeway indicates that fields were most probably built separately and could therefore have different ages. However, the OSL ages show that the raised fields were in use during a rather short time span, about 100-200 years at the beginning of the 2nd millennium. This is consistent with the raised fields in Bermeo (Rodrigues et al., 2015), where fields were used for short periods, and with the relatively brief occupation of settlements associated to raised fields in the northern

LM close to Santa Ana (Walker, 2004).The age below the infilling of Canal 2, BC 2870-2130, is consistent with the age obtained for the base of the ridge profile (BC 3500-2680), indicating that the sediments below 60 cm in Canal 2 were not reworked for the construction of the raised fields.

The suggested original depth of Canal 2 is consistent with these results and it may be reasonably assumed that the canal was originally 50 cm deeper (Fig.11). Hence, the difference between the canal and the ridge was at least 150 cm, without taking

into account the eroded material from the ridge (Fig.11). Most probably all fields were much steeper, as in the case of Bermeo (Rodrigues et al., 2015). Compared to the platform fields in the northern part of the Llanos de Moxos (Walker, 2004; Lombardo, 2010), the large fields of San Borja are about three times higher. In addition, the raised fields in this area are associated to causeways which most probably were needed to reach the fields during high water season. It is important to note that causeways do not embank the fields, providing open runoff. There is no evidence that causeways were built to hold back

water in the raised field's area or to extend the time of flooding in the area, as has been proposed for canals in the Apere Region (Erickson and Walker, 2009). During the year 2012, the canals adjoining the causeways were already completely dried out in July. Most causeways found in the adjacent settlement area, away from raised fields, must have served as a form of transportation and communication between settlements during the wet season (Erickson and Walker, 2009), suggesting that even the most elevated parts got flooded (Fig. 3). Because of its location close to the Andean Piedmont, San Borja gets on

average 400 mm more precipitation compared to the northern LM (Hijmanns et al., 2005) and ground water levels during the dry season are 2 to 3 m below the surface, up to 3 times higher compared to the northern LM (Hanagarth, 1993). Because of this, during the rainy season sediments get saturated quickly. This combination makes flooding in the southern LM much more pronounced compared to the northern part of the LM and may be an important reasons why fields in the southern part of the LM are commonly much higher.

**5.2.2 Radiocarbon dating $^{14}$C**

Charcoal derived from the CW (BC 765-471; BC 566-998) and from the ridge profile of Field 2 (BC 1438-1262) are much older than the construction of the fields, probably predating the occupation of the area. While in the CW charcoal was plentiful, in the area of raised fields only one single piece has been found. In comparison, charcoal in the raised fields of Bermeo was more abundant, found in the canals and specific layers in the elevated fields (Rodrigues et al., 2015). In Bermeo the raised

fields are under dense forest and it has been shown that the fields were in use during at least two different periods. This suggests that fire could have been used to clear the forest between periods of field use. The use of fire for raised field management in the northern LM has been also suggested by Whitney et al., 2014; Erickson and Balée, 2006. The use of fire however stands



in contrast to the fact that charcoal was not directly found in the fields in San Borja and there is no evidence of the fields having been used during several different periods and managed with fire.

The age of the Palaeosol I (cal BC 5212- 4940) below Field 2 is consistent with the age of the sediments that cover it (BC 3500-2680) and with the general chronology. There is an important difference between the ages from the two fractions of

5 Palaeosol II in the NF (cal BC 5934-5775 for the humin fraction and cal BC 976-816 humate fraction), this might point towards significant contamination. In theory, the humin fraction (residues) is considered to be more stable and represents the oldest age, whereas the humate (humic acid) fraction gives crucial information about the degree of contamination and, consequently, its reliability (Pessenda et al., 2001). The fact that the sample in the Palaeosol II of the NF was taken relatively near the surface (at a depth of 65 cm) could point towards contamination of the humate fraction with modern carbon from the surface (Walker,

2005). In addition, the fact that the Palaeosol I is chronologically consistent with the OSL ages above it, suggests that the age of the humine fraction might be more reliable. Taking the humin ages of the two palaesols does suggest that these two could have belonged to the same palaeosurface. Thus, the topography was much steeper than today and the depressions were later filled up with sediments. Palaeosol I could have been covered by a crevasse splay, while Palaeosol II could have been covered more slowly, with fine flood sediments from the surrounding area or the overflow of a distal river.

**5.3 Local soil properties versus raised fields**

Hydromorphic characteristics present in all raised fields shows that they are highly influenced by high water tables. The average depth of the water table can be derived from the Fe/Al ratio (McQueen, 2006), showing the in situ accumulation of iron which in some profiles is clearly expressed with a peak. Field observations shows that for most profiles manganese tends to accumulate some centimetres above the iron oxides. This seems to be common in hydromorphic soils, because of the

20 different solubility of iron and manganese (Lindbo et al., 2010). The depth of these oxides and the Fe/Al ratio suggest that the depth of the modern water table in the ridge profiles of Fields 1 and 2 normally oscillates between 40 cm and 50 cm below the surface, while in the lower Fields 3 and 4 the depth of the water table is between 30 and 35 cm respectively. The hydromorphic features formed at the time of field construction have probably been erased, as hydromorphic features are forming continuously and the fields were not used for a very long period of time. As already mentioned, no connection exists between the height of

25 the raised fields and the topography. For raised fields studied in the northern part of the LM it has been shown that these were built higher on finer sediments, as coarse sediments provided better drainage (Rodrigues et al., 2016). Surprisingly, in San Borja the opposite can be observed: smaller fields are built on finer sediments and the higher fields are built on the coarser sediments. There are several possible explanations for this apparent contradiction.

When comparing the relative depth of the water table below the ridge in the high fields vs the low fields, we can see that there

is a small difference of about 10-20 cm. Surprisingly, the water table below the ridge in Field 3, which has a similar height but much coarser sediments than Field 4, is almost at the same depth as in Field 4. This suggests that the drainage is not significantly better in the sandy area. This might be explained by the fact that the regional water table in San Borja is very close to the surface (Hanagarth, 1993; Miguez-Macho and Fan, 2012), hindering vertical drainage. Furthermore, as seen in





figure 6, the sandy areas on which the fields were built seem to be enclosed by fine sediments, hence also hindering lateral water movement.

Besides the hydrology, another important factor determining the height of the fields seems to be the sedimentary characteristics of each field. In the case of the lower/smaller fields, sediments become finer towards the bottom, with a high percentage of
clay in the lower layers (Fig. 4). The sediments in these deeper layers have a similar clay content (up to 28%) to the sediments in the 'no fields area' (NF). Such soils are generally avoided in agriculture because of their poor physical properties (e.g. low permeability and poor soil structure) and workability. It is therefore conceivable that due to the limited availability of coarser sediments in these deeper layers only the uppermost layers were used to raise the fields. In contrast, in the area of the large fields this impediment does not exist and fields could be built higher. In addition, coarser sediments are much easier to work,
which could also explain why these fields are higher.

Taking into account the very different soil properties and fertility status of the sediments, people could have brought the more sandy sediments from the large fields area in order to improve the soil conditions of the area with clay rich sediments, and vice versa. However, there is no evidence of this. On the contrary, as we can see in Field 3, the northern part of the field is composed of sandy loams, whilst further south the field is formed by silty loams and there is no evidence of attempting to improve the
field's soil structure by mixing the finer and coarser sediments (Fig. 6).

In general, the geochemical and physical properties of the soils here are similar to those of other soils studied along the Maniqui (Guèze et al., 2013). This area has unfertile, acidic sandy-loamy soils. The exceptional high pH of Palaeosol II in profile NF is a result of the considerably higher amount of Ca and Na within the same layer (Fig. 6 and Tab.1). Such saline soils, with accumulation of carbonates in the subsoil, have been described by other authors (Boixadera et al., 2003; Hanagarth, 1993).
The Ca and Na could have been supplied through capillary rise (Boixadera et al., 2003). Similarly, as shown by Boixadera (2003), the present profile is poorly drained and clay-rich. The layer just above Palaeosol II, with considerably more clay, could further prevent the outwash of the bases. If the Ca and Na come from capillary rise or are relict features from Palaeosol II, is beyond the scope of this study.

In general there is a clear relationship between particle size and $CEC_{eff}$. As expected, soil fertility in Fields 2 and 3, which have
been built on sand, is extremely low. Due to their limited capacity to retain cations and their high water conductivity, sandy soils tend to quickly leach valuable nutrients like calcium and magnesium and, as a result, plant growth might be hindered (Schachtschabel et al., 2002). This is reflected in the elemental composition of Ridge 3, where the more soluble elements CaO, Na, $K_2O$ and MgO are washed out from the upper parts of the profiles, showing higher values at the bottom (Fig. 8). The soil's acidity further increases the amount of exchangeable $Al^{3+}$ and Mn, which can be toxic for most plants (Jones, 2012; Cronan
and Grigal, 1995). Fields 1 and 4 have much finer sediments and are able to retain more cations, making them more fertile. Besides $CEC_{eff}$ $P_{(av)}$ is one of the most important limiting nutrients for crops (Fageria et al., 2011). In general, $P_{(av)}$ values in all profiles are very low to low, with less than 5 ppm – 10 ppm, constituting <0.5% of the total amount of phosphorous. Similar values (4.5 to 10 ppm), comprising the same region, were reported by Guèze et al. (2013). Low $P_{(av)}$ values seem to be common in acidic soils as phosphorous normally occurs bounded to other elements and is therefore not directly available to plants





(Schachtschabel et al., 2002). The results, however, also show some quit high values of $P_{av} > 20ppm$, which are related to the higher levels of total amount of P. The higher P values are related to the Fe/Al, which coincides with the average depth of the water table. Similar positive links between $P_2O_5$ and Fe/Al have been reported by other studies (Lopez et al. 2005, 2006; Huang et al. 2005). The overall high values of total $PO_4$ can be explained by the relatively young age of the sediments and the fact

that sediments coming from the Andes are high in phosphorous. Up to 2200 ppm (for $P_2O_5$) and up to 4300 ppm (for CaO) have been reported for river sediments coming from the Andes (Guyot, 1992). While CaO, Mn, MgO are leached relatively fast, while P in acidic conditions is stabilized in the soil by iron and aluminium.(Schachtschabel et al., 2002). Due to the seasonal floods and droughts, oxides (e.g. $P_2O_5$ and $Fe_2O_3$), along with other elements, are redistributed and accumulated in the soil (Scott 2009). It seems, therefore, unlikely that the accumulation of P in specific layers is the result of an anthropogenic

enrichment of the soil due to practices of intensive manuring, as reported for other anthropogenic soils (Holliday, 2004; Costa et al., 2013; Glaser and Woods, 2004).

In the ridge profile of the coarser Field 3, the leaching of cations, geochemical changes and enhanced hydromophism all suggests that, in the long run, the construction of raised fields could accelerate soil weathering. These processes should be taken into account when considering raised fields as a model for sustainable agriculture today.

It has been suggested that manure or muck grown in the canal could have been used to fertilize the fields, allowing continuous production without the need of fallow periods (Erickson, 1994; Lee, 1997; Barba, 2003; Saavedra, 2006). Nevertheless, in order to produce green manure the canals would have had to retain water during the dry season, which is not the case in the San Borja fields nor in other fields studied in the LM (Lombardo et al., 2011a; Rodrigues et al., 2015). It is possible, however, that earth from the canals could have been reused to raise the fields. If we compare the soil properties from the infilling of the

canal and the raised field, the former has slightly higher content of organic matter, $P_{av}$ and $CEC_{eff}$ than the latter. By adding sediment from the canal onto the elevated bed they could have improved the field's soil fertility. However, it has to be considered that the fertility of the soil from the canal that we see today is the result of at least 500 years of accumulation of sediments and organic matter. If, in pre-Columbian times, the fields had been in use continuously, there would not have been sufficient fertile sediments in the canals left to fertilize the fields after one growing season and manure would have had to been

brought from somewhere else.

Up till now raised fields studied in the LM have not shown evidence of intensive manuring but rather of extensive agricultural practices (Lombardo, 2010; Lombardo et al., 2011a; Rodrigues et al., 2015; Rodrigues et al., 2016). However comparing the fertility status of the raised fields studied here with the ones in the northern LM, the results show that the soils in San Borja are considerably more fertile.

The soils in the northern LM are much older and much more weathered (Rodrigues et al. 2015.). On the other hand, density of fields in the northern LM is much higher than in San Borja. This high density of raised fields has been interpreted as the result of intermittent use by small groups of mobile people which were shifting their fields over a period of hundreds of years (Rodrigues et al., 2016).



Even though the fertility status of the soils in San Borja is better, raised fields here would similarly have needed fallow periods, especially those built on the sandy sediments. It is probable that the fields built on the more fertile soils could have been cultivated for longer periods, with shorter fallows. If we assume a similar scenario of small groups of mobile people in the present study area, the better soil quality could explain the much lower density of raised fields here compared to the northern

LM.

Another reason for the lower density could be related to the widespread availability of more elevated well drained areas. These areas normally do not get flooded and similarly, as shown for the region of Bermeo, raised fields could have been constructed to overcome periods of more severe and frequent flooding (Lombardo et al., 2011a; Rodrigues et al., 2015). On the contrary, the northern LM, is affected by ponding water on a regular basis, because of relatively impermeable soils, and raised fields

were needed to improve the drainage (Rodrigues et al., 2016). Consequently, the density of fields in the different parts of the LM could also be related to the frequency of use, with raised fields in the southern LM used only during periods of extreme events while in the northern part fields were used annually.

However, as almost no data exists about the size and timing of pre-Columbian occupations such scenarios are difficult to prove. Furthermore, as argued by McKey and Rostain (2014), the raised fields might have very possibly been complemented by other

subsistence systems which also could have been different for each region.

Nevertheless, information about soil properties are important in order to understand the development of societies (McNeill and Winiwarter, 2006), differences in agricultural strategies and in distribution and density of people living in the LM. Up till now, raised fields studied in the LM all show that fields were constructed with the main purpose of drainage. The fact that some of the raised fields studied here were constructed on the highly unfertile sands supports the idea that drainage was the first priority.

There is no clear evidence suggesting that raised fields were more productive compared with similar soils which are naturally drained. Hence, there is no indication which suggests that the construction of raised fields was a highly productive strategy which could sustain dense populations.

As already proposed with the first description of raised fields in 1916 by Erland Nordenskiöld (Denevan, 2009), they must have played a crucial role in protecting the crops from the floods. The abundant causeways, even on the more elevated area,

further suggest that in San Borja flooding used to be a frequent problem.

As similarly suggested for the raised fields in Bermeo, the period of use in San Borja coincides with a period of higher ENSO activity, which has been reported to be an important factor responsible for extreme floods and droughts during the past 2500 years in South America (Meggers, 1994; Markgraf and Díaz, 2000; Moy et al., 2002; Rein et al., 2005). Some major floods in the LM have been associated with the negative ENSO phase La Niña, where rainfall is above normal in the Basin (Aalto et al.,

2003). Moy et al. (2002) reported higher frequency of extreme ENSO events occurring between 1000 and 2000 years ago, with its maximum around AD 800, which coincides with the time when fields were in use. However, as reported for the extreme event in 2014, severe flooding can as well occur in absence of ENSO, as a result of tropical and subtropical changes in South Atlantic Sea Surface Temperature (SST) (Espinoza et al., 2014). Today losses of harvest due to flooding are frequently reported in the LM (UNDP, 2011).





## 6 Conclusion

We analyse raised fields of different sizes which were built in an area, near San Borja, with a complex fluvial history. Different generations of palaeo-rivers, partly overlapping each other, coexist in the area, resulting in a heterogeneous depositional environment. This is reflected in the great variability of sediment particle size of the excavated raised fields. The results show

that differences in field size and height are the result of an adaptation to this heterogeneous depositional environment. The dimension of the fields is related to particle size. Only coarse, silty to sandy sediments were used for the construction of the raised fields. The height of the fields depends on how deep the coarse sediments are: fields are relatively small where the coarse sediments are limited to the surface, whilst in areas where the subsoil was also made of coarse sediments these could be used to build larger fields. Areas with exclusively fine clay rich sediments were not used for the construction of raised

fields. Raised fields were built by piling up the sediments taken from the excavation of the adjacent canal; there is no evidence of other agricultural strategies such as mixing of sediments or intensive manuring. Geochemical changes along the stratigraphic profiles show that the construction of fields might accelerate the weathering process on the long term, calling into question the idea of reintroducing raised fields as a very productive model of sustainable agriculture for today. The raised fields in the area are always associated to causeways. There is no evidence that causeways were built to manage the floodwaters; they were

more likely used to reach the fields and to connect settlement areas. Although the construction of raised fields did not directly improve soil fertility, leading to higher productivity, it was of major importance to protect crops during severe flooding.

## Author Contribution

L.Rodigues and U. Lombardo conceived and designed the study. L.Rodrigues, U. Lombardo, P.Huber and H. Veit performed field work. L. Rodrigues, P.Huber and S.Mohr carried out laboratory analyses. M.Trauerstein conducted OSL measurements.

H. Veit secured funding. L.Rodrigues prepared the manuscript with contributions from all co-authors.

## Acknowledgements

The present study has been funded by the Swiss National Science Foundation (SNSF), grant no SNF 200020-141277/1, and performed under authorisation N_ 017/2012 issued by the Unidad de Arqueología y Museos (UDAM) del Estado Plurinacional de Bolivia. We thank Dr. M.R. Michel López from the Ministerio de Culturas and our Bolivian counterpart Dr. J.M. Capriles

for their support. A special thanks to the owners of the farms El Progresso and Campo España and their workers for their logistical support in the field and for allowing us free access to their land. Fieldwork assistance by B. Vogt, L.M. Salazar and C. Welker is gratefully acknowledged. We thank Dr. D. Fischer for technical support in the laboratory. X-ray fluorescence spectroscopy (XRF) was measured at the Geological Institute of the University of Fribourg. A special thanks to E. Canal for improvement of the manuscript.



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




**Table 1 Down-profile values of selected geochemical parameters and grain size of all Fields (ridges and canals).**

| Soil geochemical and physical properties | | | | | | | | | | | | |
|---|---|---|---|---|---|---|---|---|---|---|---|---|
| | Sand | Silt | Clay | pH | Corg | N | CECeff | Bs | Ca | Al | Ca/Al | Pav |
| Depth: | 63µm- | 4µm-63µm | <4.00µm | | % | % | mmolc/kg | % | % | % | | ppm |
| **Ridge 1** | | | | | | | | | | | | |
| 0-10 | 52.10 | 39.73 | 8.17 | 4.1 | 0.68 | 0.08 | 19 | 54 | 32 | 39 | 0.82 | 3.7 |
| 10-20 | 44.58 | 44.97 | 10.46 | 4.1 | 0.4 | 0.03 | 20 | 40 | 20 | 54 | 0.37 | 1.0 |
| 20-30 | 47.42 | 42.97 | 9.62 | 3.9 | 0.23 | 0.03 | 18 | 65 | 30 | 31 | 0.98 | 1.2 |
| 30-40 | 49.72 | 41.44 | 8.84 | 4.2 | 0.3 | 0.034 | 26 | 72 | 34 | 26 | 1.30 | 0 |
| 40-50 | 45.78 | 42.65 | 11.58 | 4.4 | 0.22 | 0.03 | 27 | 84 | 46 | 14 | 3.40 | 0.2 |
| 50-60 | 41.36 | 47.14 | 11.51 | 4.4 | 0.13 | 0.02 | 29 | 82 | 46 | 17 | 2.75 | 12 |
| 60-70 | 37.99 | 49.23 | 12.77 | 4.3 | 0.14 | 0.03 | 46 | 92 | 40 | 8 | 5.30 | 18.7 |
| 70-80 | 43.04 | 45.42 | 11.54 | 4.4 | - | - | 100 | 96 | 53 | 3 | 15.98 | 16.6 |
| 80-90 | 37.79 | 50.18 | 12.04 | 4.5 | 0.19 | 0.03 | 67 | 95 | 37 | 4 | 8.47 | 20.6 |
| 90-100 | 27.16 | 57.09 | 15.76 | 4.5 | 0.16 | 0.03 | 56 | 100 | 39 | 0.00 | #DIV/0! | - |
| **Canal 1** | | | | | | | | | | | | |
| 0-10 | 43.94 | 47.41 | 8.65 | 3.9 | 0.87 | 0.10 | 30 | 60 | 46 | 6 | 1.36 | 3.4 |
| 10-20 | 43.69 | 46.21 | 10.11 | 3.9 | 0.59 | 0.06 | 30 | 58 | 43 | 4 | 1.14 | 1.2 |
| 20-30 | 39.07 | 48.59 | 12.34 | 4.1 | 0.49 | 0.05 | 4 | 55 | 35 | 2 | 0.83 | 2.3 |
| 30-40 | 30.91 | 55.35 | 13.74 | 4 | 0.30 | 0.04 | 45 | 62 | 39 | 4 | 1.05 | 6.8 |
| 40-50 | 19.71 | 62.59 | 17.70 | 4.2 | 0.19 | 0.04 | 71 | 70 | 38 | 4 | 1.29 | 0.5 |
| 50-60 | 33.49 | 53.97 | 12.55 | 4.2 | 0.14 | 0.03 | 52 | 88 | 42 | 0.3 | 3.53 | 8.5 |
| 80-90 | 46.47 | 44.87 | 8.66 | 4.3 | 0.08 | 0.02 | 52 | 93 | 32 | 0.2 | 4.30 | 15.2 |
| 90-100 | 45.47 | 44.85 | 9.68 | 4.6 | 0.10 | 0.02 | 74 | 96 | 34 | 0.3 | 9.06 | 26.8 |
| **Ridge 2** | | | | | | | | | | | | |
| 0-10 | 65.60 | 27.55 | 6.85 | 4.2 | 0.74 | 0.07 | 17 | 42 | 30 | 55 | 0.55 | - |
| 10-20 | 59.46 | 31.91 | 8.63 | 4.0 | 0.46 | 0.05 | 16 | 19 | 12 | 79 | 0.15 | - |
| 20-30 | 58.71 | 31.38 | 9.92 | 4.0 | 0.33 | 0.03 | 15 | 17 | 10 | 81 | 0.12 | - |
| 30-40 | 56.82 | 34.13 | 9.04 | 4.0 | 0.22 | 0.02 | 12 | 15 | 8 | 83 | 0.10 | - |
| 40-50 | 60.49 | 32.15 | 7.36 | 4.0 | 0.16 | 0.02 | 11 | 18 | 11 | 80 | 0.14 | - |
| 60-70 | 58.34 | 32.58 | 9.08 | 4.0 | 0.13 | 0.02 | 12 | 17 | 10 | 82 | 0.13 | - |
| 70-80 | 61.42 | 29.65 | 8.94 | 4.1 | 0.11 | 0.01 | 11 | 22 | 13 | 76 | 0.18 | - |
| 90-100 | 60.77 | 31.65 | 7.58 | 4.0 | 0.10 | 0.01 | 16 | 17 | 9 | 82 | 0.11 | - |
| 100-110 | 73.96 | 20.54 | 5.50 | 4.0 | 0.09 | 0.01 | 16 | 12 | 6 | 83 | 0.07 | - |
| 110-120 | 70.79 | 22.07 | 7.15 | 4.0 | 0.05 | 0.01 | 17 | 16 | 9 | 85 | 0.10 | - |
| 120-130 | 77.98 | 17.17 | 4.85 | 3.1 | 0.06 | DL | - | - | - | - | - | - |
| 130-140 | 84.61 | 12.21 | 3.18 | 4.0 | 0.04 | DL | - | - | - | - | - | - |
| 140-50 | 90.88 | 7.53 | 1.59 | 4.1 | 0.03 | DL | - | - | - | - | - | - |
| **Canal 2** | | | | | | | | | | | | |
| 0-10 | 43.93 | 39.41 | 16.66 | 4.0 | 1.13 | 0.11 | 20 | 33 | 9 | 66 | 0.14 | - |
| 15-25 | 37.08 | 49.54 | 13.38 | 4.0 | 0.39 | 0.04 | 23 | 38 | 13 | 60 | 0.22 | - |
| 30-40 | 50.00 | 36.70 | 13.31 | 4.1 | 0.30 | 0.03 | 25 | 23 | 8 | 76 | 0.10 | - |
| 40-50 | 41.37 | 41.55 | 17.08 | 4.0 | 0.24 | 0.03 | 33 | 25 | 10 | 74 | 0.13 | - |
| 50-60 | 46.85 | 37.63 | 15.52 | 3.9 | 0.21 | 0.03 | 42 | 22 | 7 | 77 | 0.10 | - |
| 60-70 | 63.81 | 25.98 | 10.21 | 4.0 | 0.01 | 0.02 | 31 | 26 | 11 | 74 | 0.15 | - |
| 70-80 | 55.77 | 32.88 | 11.35 | 4.0 | 0.10 | 0.01 | 31 | 31 | 16 | 68 | 0.23 | - |
| 80-90 | 50.84 | 37.44 | 11.72 | 4.0 | 0.08 | 0.01 | 22 | 40 | 21 | 60 | 0.36 | - |
| 100-110 | 59.80 | 30.08 | 10.13 | 4.0 | 0.06 | 0.01 | 25 | 44 | 24 | 56 | 0.42 | - |
| 110-120 | 67.82 | 25.56 | 6.63 | 4.0 | 0.07 | 0.01 | 22 | 39 | 15 | 6 | 0.25 | - |



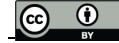

| Depth: | Sand 63µm- | Silt 4µm-63µm | Clay <4.00µm | pH | Corg % | N % | CECeff mmolc/kg | Bs % | Ca % | Al % | Ca/Al | Pav ppm |
|---|---|---|---|---|---|---|---|---|---|---|---|---|
| **Ridge 3** | | | | | | | | | | | | |
| 10-25 | 65.75 | 29.77 | 4.47 | 4.0 | 0.37 | 0.05 | 12 | 13 | 6 | 87 | 0.07 | 1.16 |
| 30-40 | 65.80 | 29.96 | 4.15 | 4.0 | 0.14 | 0.02 | 11 | 20 | 13 | 77 | 0.17 | 0 |
| 40-50 | 68.94 | 27.12 | 3.94 | 4.1 | 0.13 | 0.02 | 10 | 24 | 13 | 73 | 0.18 | 0 |
| 50-60 | 70.06 | 26.21 | 3.73 | 4.0 | 0.10 | 0.02 | 12 | 17 | 9 | 80 | 0.11 | 0 |
| 60-70 | 73.48 | 23.07 | 3.45 | 4.0 | 0.08 | 0.02 | 14 | 18 | 9 | 80 | 0.11 | 0.84 |
| 70-80 | 76.51 | 21.01 | 2.48 | 4.0 | 0.06 | 0.01 | 12 | 28 | 17 | 71 | 0.23 | 2.6 |
| 90-100 | 88.02 | 10.71 | 1.27 | 4.0 | 0.04 | 0.01 | 11 | 18 | 9 | 81 | 0.11 | 2.56 |
| 100-110 | 88.59 | 10.12 | 1.29 | 4.1 | 0.03 | 0.01 | 12 | 22 | 11 | 77 | 0.14 | - |
| 120-130 | 89.02 | 9.65 | 1.33 | 4.1 | 0.02 | 0.01 | 13 | 32 | 18 | 68 | 0.26 | - |
| **Canal 3** | | | | | | | | | | | | |
| 15-25 | 53.5 | 40.22 | 6.28 | 3.9 | 0.60 | 0.07 | 20 | 26 | 12 | 71 | 0.16 | 5.32 |
| 25-35 | 58.32 | 36.11 | 5.58 | 4.0 | 0.35 | 0.04 | 18 | 40 | 17 | 56 | 0.30 | 10.48 |
| 35-45 | 64.56 | 31.02 | 4.42 | 4.0 | 0.03 | 0.03 | 20 | 34 | 11 | 63 | 0.18 | 9.8 |
| 45-55 | 67.62 | 28.39 | 3.99 | 4.0 | 0.18 | 0.03 | 15 | 32 | 10 | 65 | 0.15 | 12.6 |
| 55-65 | 79.50 | 18.32 | 2.18 | 4.0 | 0.06 | 0.01 | 13 | 32 | 7 | 67 | 0.10 | 20.96 |
| 65-75 | 82.20 | 15.99 | 1.81 | 4.0 | 0.05 | 0.01 | 13 | 23 | 8 | 77 | 0.10 | 17.6 |
| 75-85 | 89.06 | 9.92 | 1.01 | 4.0 | 0.03 | 0.01 | 18 | 21 | 8 | 78 | 0.10 | 14.8 |
| 85-90 | 80.06 | 17.46 | 2.48 | 4.0 | 0.04 | 0.01 | 12 | 21 | 6 | 79 | 0.08 | 11.2 |
| 90-100 | 90.80 | 8.44 | 0.77 | 4.0 | 0.02 | 0.01 | 14 | 16 | 4 | 84 | 0.05 | 7.6 |
| **Ridge 4** | | | | | | | | | | | | |
| 0-10 | 41.40 | 49.59 | 10.72 | 4 | 0.75 | 0.08 | 56 | 71 | 35 | 26 | 1.33 | 2.04 |
| 10-20 | 36.01 | 52.88 | 11.34 | 4.3 | 0.32 | 0.04 | 34 | 83 | 39 | 15 | 2.60 | 0 |
| 20-35 | 36.24 | 49.87 | 15.80 | 4.2 | 0.34 | 0.05 | 71 | 88 | 37 | 10 | 3.84 | 0.04 |
| 40-50 | 35.29 | 49.05 | 16.91 | 4.6 | 0.28 | 0.04 | 80 | 97 | 42 | 3 | 16.04 | 1.2 |
| 50-60 | 34.98 | 46.93 | 19.05 | 4.8 | 0.25 | 0.05 | 98 | 100 | 45 | 0.00 | #DIV/0! | 1.4 |
| 70-80 | 21.47 | 59.23 | 21.53 | 5.1 | 0.21 | 0.05 | 128 | 100 | 45 | 0.00 | #DIV/0! | 6.2 |
| 80-90 | 2.23 | 71.51 | 26.26 | 5.2 | 0.21 | 0.05 | 75 | 100 | 87 | 0.00 | #DIV/0! | 9.04 |
| 90-100 | 6.02 | 68.50 | 26.60 | 5.1 | 0.19 | 0.05 | 172 | 100 | 49 | 0.00 | #DIV/0! | 6.24 |
| **Canal 4** | | | | | | | | | | | | |
| 0-10 | 29.51 | 52.58 | 19.02 | 4 | 0.86 | 0.10 | 52 | 75 | 32 | 22 | 1.43 | 5.04 |
| 10-20 | 28.79 | 55.02 | 16.73 | 3.9 | 0.40 | 0.05 | 36 | 67 | 36 | 30 | 1.19 | 3.6 |
| 20-30 | 47.54 | 46.29 | 14.41 | 4 | 0.33 | 0.05 | 37 | 72 | 35 | 25 | 1.39 | 4.6 |
| 30-40 | 22.73 | 57.60 | 19.67 | 4 | 0.52 | 0.06 | 85 | 80 | 32 | 18 | 1.73 | 3.56 |
| 50-60 | 21.03 | 59.42 | 20.22 | 4.2 | 0.29 | 0.04 | 84 | 87 | 37 | 12 | 3.14 | 4.04 |
| 70-80 | 4.34 | 67.83 | 28.68 | 4.7 | 0.20 | 0.06 | 170 | 100 | 42 | 0.00 | #DIV/0! | 4.48 |
| **No Field** | | | | | | | | | | | | |
| 0-10 | 18.09 | 62.02 | 19.89 | 3.9 | 1.13 | 0.13 | 32 | 67 | 31 | 28 | 1.11 | 2.6 |
| 10-20 | 15.70 | 62.65 | 21.65 | 4.0 | 0.70 | 0.09 | 38 | 66 | 29 | 32 | 0.92 | 0.88 |
| 20-30 | 16.41 | 60.18 | 23.41 | 4.0 | 0.52 | 0.08 | 75 | 43 | 19 | 56 | 0.34 | 0.8 |
| 30-40 | 22.42 | 56.75 | 20.83 | 4.1 | 0.38 | 0.07 | 82 | 47 | 20 | 52 | 0.39 | 0 |
| 40-50 | 39.24 | 46.42 | 14.33 | 4.2 | 0.31 | 0.06 | 75 | 52 | 22 | 48 | 0.46 | 0 |
| 50-60 | 4.12 | 63.34 | 32.54 | 4.3 | 0.20 | 0.04 | 51 | 63 | 25 | 37 | 0.69 | 0 |
| 60-70 | 13.24 | 65.31 | 21.45 | 7.9 | 0.34 | 0.06 | 121 | 100 | 50 | 0.00 | #DIV/0! | 0 |
| 70-80 | 11.55 | 65.47 | 23.02 | 4.7 | 0.23 | 0.05 | 71 | 97 | 42 | 8 | 14.70 | 3.8 |
| 80-90 | 5.11 | 69.30 | 25.58 | 4.7 | 0.23 | 0.06 | 110 | 100 | 43 | 0.5 | 87.41 | 2.8 |
| **Causeway** | | | | | | | | | | | | |
| 0-10 | 29.28 | 59.33 | 11.39 | - | 0.77 | 0.10 | - | - | - | - | - | 2.4 |
| 20-30 | 35.16 | 51.75 | 13.10 | - | 0.32 | 0.04 | - | - | - | - | - | 0 |
| 30-40 | 41.51 | 47.29 | 11.21 | - | 0.23 | 0.04 | - | - | - | - | - | 0 |
| 50-60 | 39.22 | 47.96 | 12.81 | - | 0.25 | 0.04 | - | - | - | - | - | 3.68 |
| 60-70 | 42.07 | 44.61 | 13.32 | - | 0.18 | 0.04 | - | - | - | - | - | 3.72 |
| 80-90 | 32.92 | 50.67 | 16.41 | - | 0.13 | 0.04 | - | - | - | - | - | 1.28 |
| 90-100 | 29.28 | 59.33 | 11.39 | - | 0.12 | 0.03 | - | - | - | - | - | 2.76 |




**Table 2** AMS radiocarbon ages of charcoal and soil samples, given both as 14C age BP and calibrated radiocarbon age in AD/BC format at two-sigma level. OSL ages are given as years before sampling (rounded to the next 5 years) and converted to AD/BC format to allow direct comparison with radiocarbon ages.

**Radiocarbon ages $^{14}$C**

| Profile | Palaeosol I Field 2 | Causeway | Causeway | Ridge 2 | Palaeosol II NF |
|---|---|---|---|---|---|
| Depth (cm) | 270 | 60 | 60 | 80 | 65 |
| Material | humin/No humates | charcoal | charcoal | charcoal | humin/humates |
| $^{14}$C age | 6163 ± 41 BP | 2590 ± 23 BP | 2451 ± 30 BP | 3139 ± 30 BP | 7034 ± 31 BP/2790 ± 28 BP |
| 95.4 % (2σ) cal age ranges | BC 5212- 4940 | BC 765-471 | BC 566-398 | BC 1438-1262 | BC 5934-5775/ BC 976-816 |
| RAUPD | 1 | 0.953 | 0.791 | 1 | 0.801/1 |
| Lab Number | D_AMS -006318 | BE-3265.1.1 | BE-3266.1.1 | D-AMS 002333 | D-AMS 006330 |
| Cal Date | 30.11.2015 | 05.02.2016 | 05.02.2016 | 05.02.2016 | 26.01.2016 |

| **OSL ages** | | | | |
|---|---|---|---|---|
| **Profile** | **Ridge 2** | **Ridge 2** | **Canal 2** | **Canal 2** |
| Depth (cm) | 40 | 140 | 36 | 60 |
| OSL age | 790 ± 70 (MAM) | 5100 ± 410 | 635 ± 55 (MAM) | 4510 ± 370 |
| Converted to A.D. | AD 1150-1290 | BC 3500-2680 | AD 1320-1430 | BC 2870-2130 |

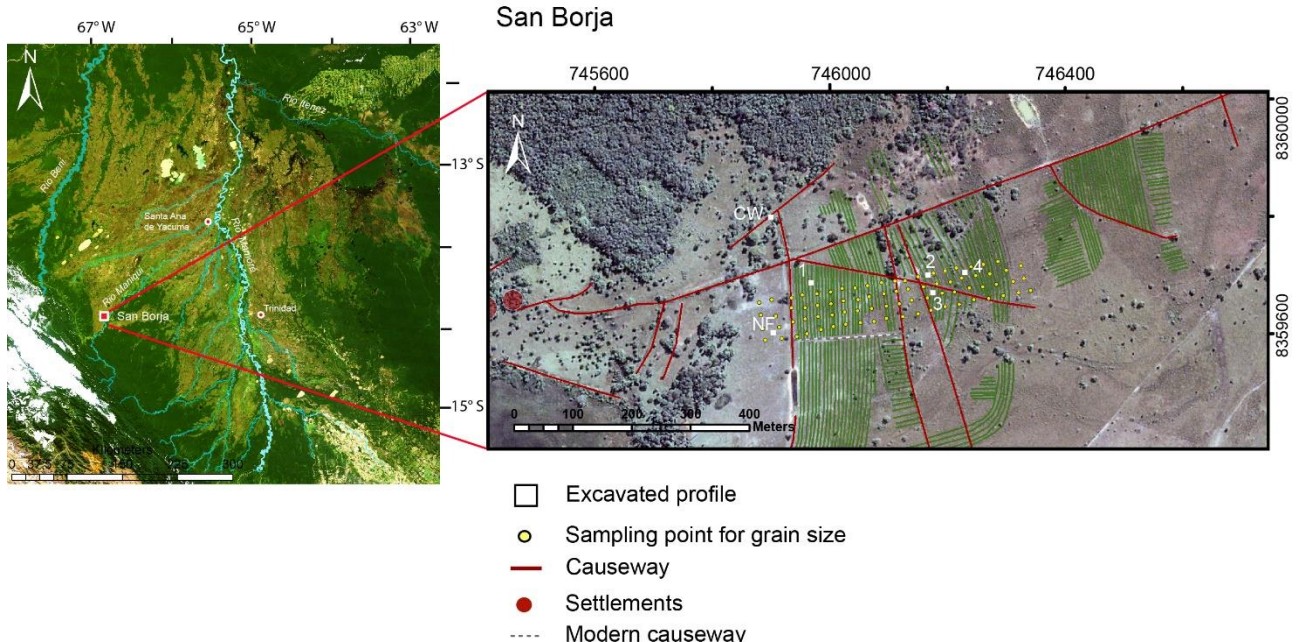

**Figure 1: Study site in the Bolivian Lowlands showing raised fields and causeways, including the location of the excavated profiles and locations sampled for grain size analysis.**





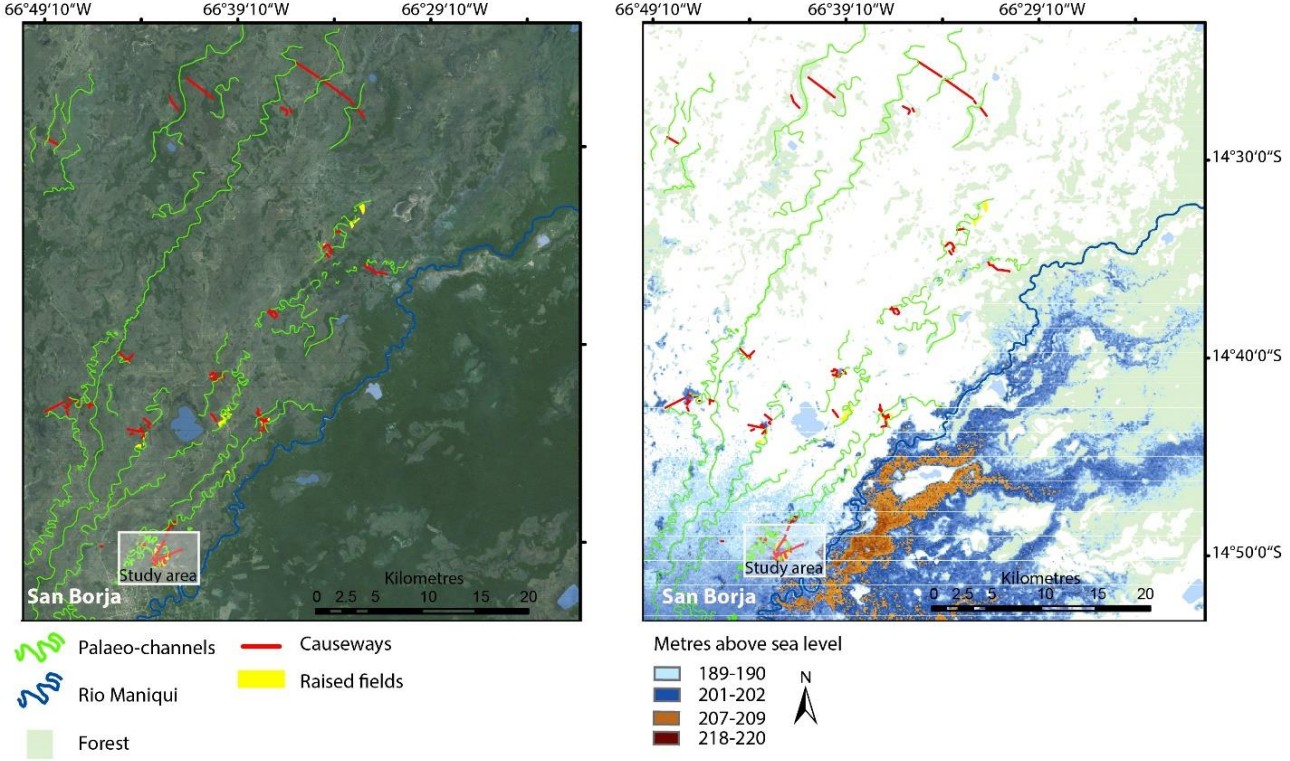

**Figure 2 Left: Google earth image showing mapped palaeo-river features, causeways (red) and raised field areas (yellow). Right: Digital elevation model (CGIAR_CSISRTM) including same mapped palaeo-river features, causeways (red) and raised field areas (yellow). Higher areas are relict sediment deposits from former rivers.**



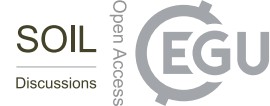

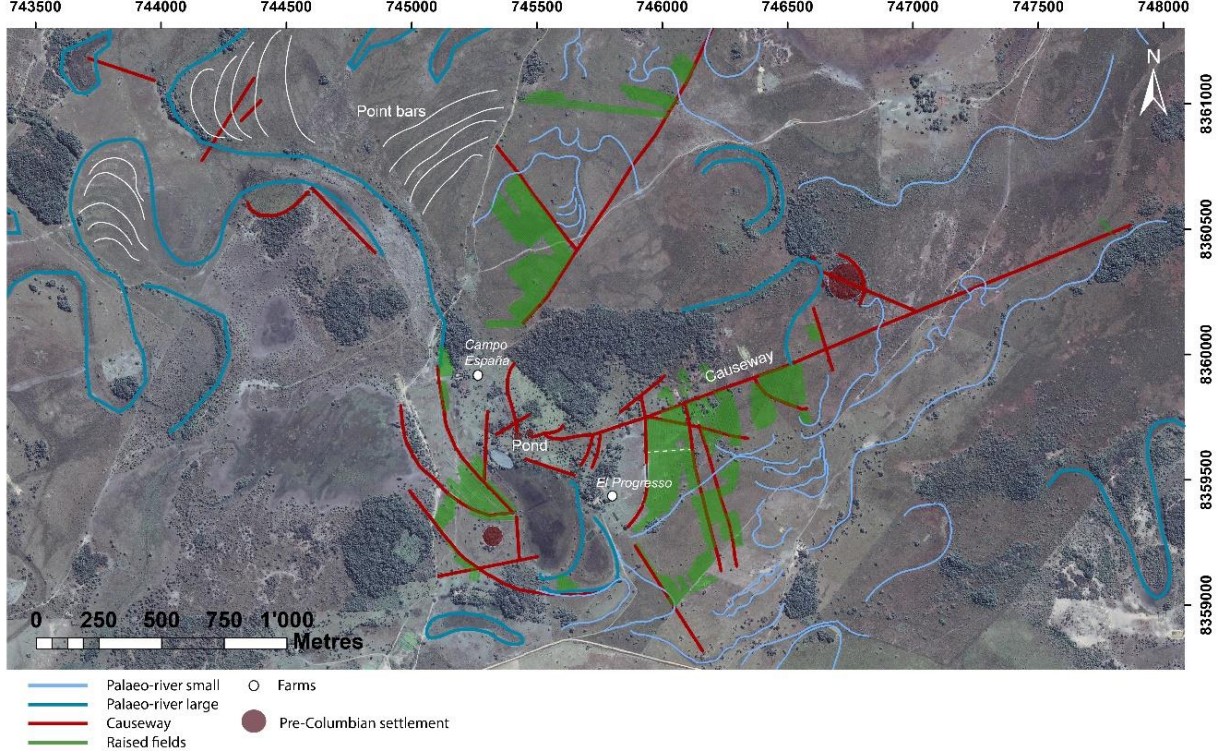

**Figure 3 Study area including anthropogenic earthworks, natural geomorphological features and the two farms Campo España and El Progresso.**



| Profile | Soil characteristics | Munsell color |
|---|---|---|
| **Ridge 1** | Silty, dull brown topsoil | 7.5 YR 6/3 |
| | Lighter dull orange layer with abundant manganese concretions 40 % (Ø 2 mm) | 7.5 YR 6/6 |
| | Orange matrix with some few iron and manganese concretions | 7.5 YR 6/4 |
| | Boundary with increasing orange iron hydromorphic mottling | 7.5 YR 6/8 |
| | Silty, hydromorphic orange mottles (30%) increasing towards the bottom | |
| | Pieces of burned earth | 7.5 YR 6/8 |
| | Water table at 110 cm (dry season 2013) | |
| **Canal 1** | Silty, brown organic rich layer | 7.5 YR 4/3 |
| | Orange mottles (5 %) and manganese concretions (Ø 2 mm) | |
| | Boundray of infilling, hydromorphic orange mottles (30 %) | 7.5 YR 4/3 |
| | Silty, hydromorphic layer, still partly saturated, reduced pale orange matrix with orange mottles | 7.5 YR 7/3 |
| | | 7.5 YR 4/3 |
| | Water table at 95 cm (dry season 2013) | |
| **Ridge 2** | Sandy loam light grey top soil | 10 YR 5/3 |
| | Hydromorphic yellow orange mottling and iron and manganese concretions (5 %) in light grey matrix | 7.5 YR 5/4 |
| | | 7.5 YR 6/6 |
| | Increasing amount of hydromorphic yellow-orange mottles and iron concretions (5 %) | 7.5 YR 6/6 |
| | Charcoal pieces | |
| | Hydromorphic yellow-orange staining of whole matrix with vertical bleaching structures | 7.5 YR 6/4 |
| | Water table below 2 m (dry season 2012) | 7.5 YR 6/4 |
| | Yellow oxidized line reffered as rust line in the text following the topography of the field present in Field 2 and 3 | |
| **Canal 2** | Greyish yellow-brown, dense topsoil | 10 YR 4/2 |
| | Slightly lighter greyish yellow brown layer with very fine manganese and iron concretions (Ø 2mm) | 10 YR 5/2 |
| | Greyish yellow brown matrix with orange mottles (20%) | 10 YR 6/2 |
| | | 10 YR 6/6 |
| | Boundary of infilling to light yellow orange matrix, abundant orange mottling (50 %) | 7.5 YR 8/3 |
| | Matrix orange 10 YR 6/4 with vertical bleaching structures (10 YR 8/3) | 7.5 YR 6/6 |
| | | 7.5 YR 8/3 |
| | Very sandy almost uniform orange layer. Water table below 2 m (dry season 2012) | 7.5 YR 6/6 |



| Profile | Soil characteristics | Munsell color |
|---|---|---|
| **Ridge 3** | Sandy loam, light grey, very thin top soil | 10 YR 5/3 |
| | Yellow-orange mottles, iron and manganese concretions (5 %) in light grey matrix | 10 YR 6/4 |
| | Increasing amount of yellow-orange mottles (30 %) | 7.5 YR 6/4 |
| | Dark brownish mottles | 7.5 YR 4/6 |
| | Dark brownish and orange mottles with vertical bleaching structures | 7.5 YR 4/6 / 7.5 YR 6/4 |
| | Very sandy uniform orange yellow layer<br>Water table below 2 m (dry season 2012) | 7.5 YR 6/4 |
| **Canal 3** | Laomy, greyish yellow-brown dense topsoil | 10 YR 4/2 |
| | Slightly lighter greyish yellow-brown with very fine manganese and iron concretions (Ø 2 mm) | 10 YR 5/2 |
| | Greyish yellow-brown matrix with orange mottles | 7.5 YR 8/3 |
| | Boundary of infilling | 7.5 YR 4/6 |
| | Light yellow-orange matrix, abundant dark brownish mottles (50 %) | |
| | Orange matrix 10 YR 6/4 with vertical bleaching structures (10 YR 8/3) | 10 YR 6/4 / 10 YR 8/3 |
| | Very sandy almost homogenous orange layer<br>Water table below 2 m (dry season 2012) | 7.5 YR 6/6 |
| **Ridge 4** | Silty, greyish brown topsoil, diffuse boundary | 7.5 YR 6/2 |
| | Silty, light brownish grey, hydromorphic mottling (5 %), manganese concretions (Ø 5 mm) | 7.5 YR 7/2 |
| | Sandy layer, hydromorphic mottling (50%) : small manganese concretions (Ø 5 mm), deformation by cattle steps (black dashed line) | |
| | Transition to light brownish-grey silty sediments, orange mottles (50%), diffuse boundary | 7.5 YR 7/2 / 2.5 YR 6/8 |
| | Increasing amount of clay and orange mottles (50%) | 5.0 YR 6/8 / 5.0 YR 8/1 |
| | Desiccation cracks (black dashed line), boundaries along cracks are filled with fine sand | 5.0 YR 6/8 |
| | Water table 100 cm (dry season 2013) | |
| **Canal 4** | Silty, brownish and organic rich topsoil | 7.5 YR 4/2 |
| | Silty, light brownish-grey matrix, manganese concretion (40 %), yellow-orange (7.5 YR 7/8) iron mottles (5 %) | 7.5 YR 7/2 |
| | Increasing amount of clay and hydromorphic orange mottles (50%) | 5.0 YR 6/8 / 5.0 YR 8/1 |
| | Silty clay, hydromorphic reddish-brown mottling 5 YR 4.8 > 50% | 5.0 YR 4.8 |
| | Water table 80 cm (dry season 2013) | |

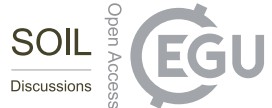

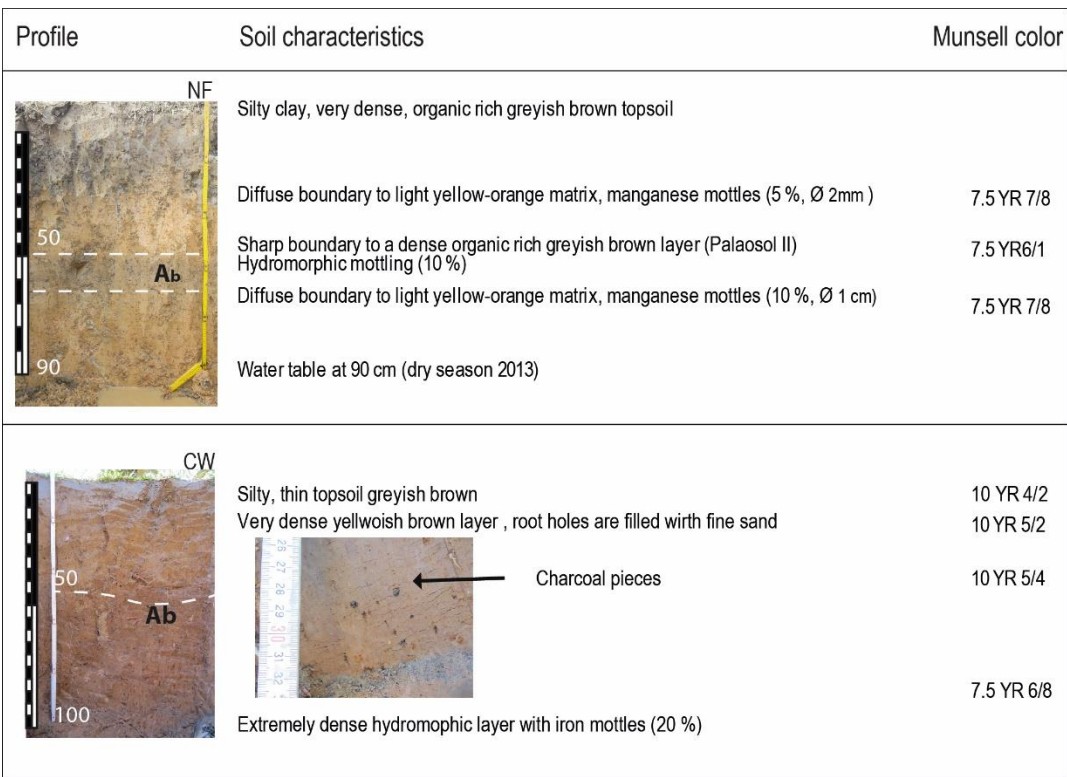

**Figure 4 Profile description: Munsell color signature is given for each layer, water table boundaries are illustrated as blue dashed lines, Ab= buried topsoil.**

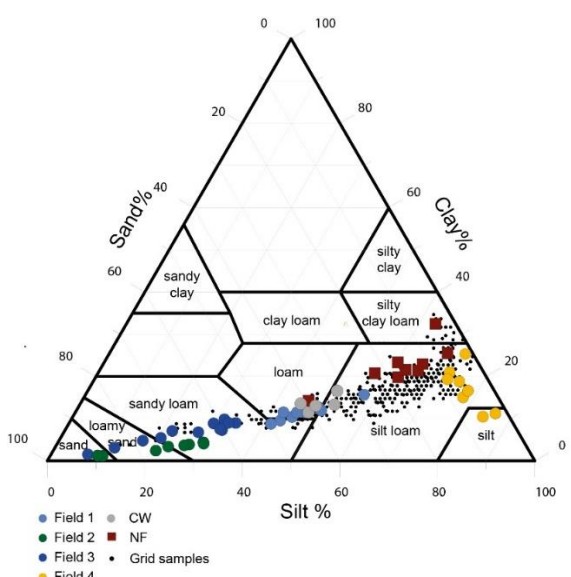

**Figure 5 Soil texture triangle including all measured samples.**





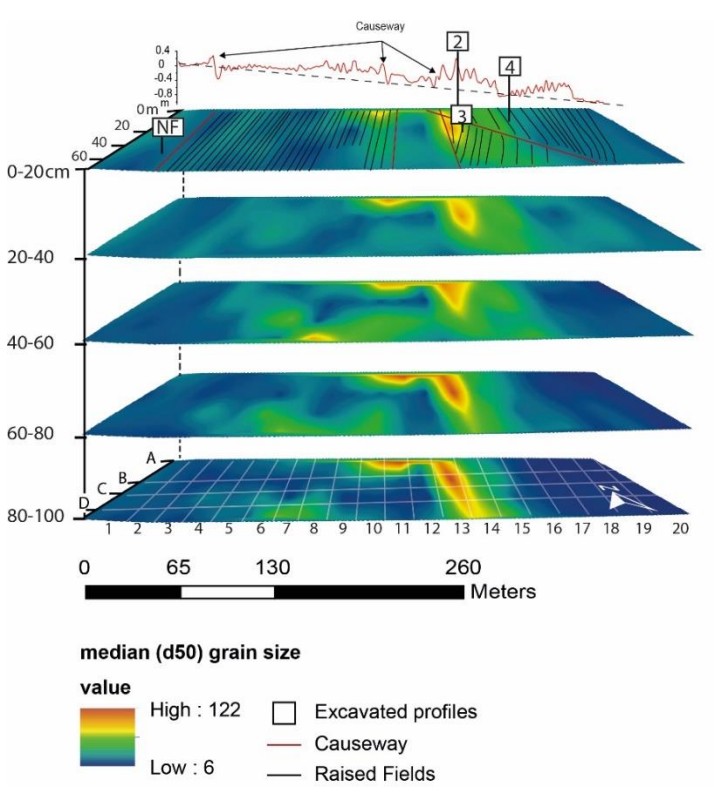

**Figure 6** Interpolated grain size distribution of the sampled area using the median particle size of each sample (d50 µm) at five different depths. Top layer includes mapped fields and excavation sites. Bottom layer shows virtual grid, samples for grain size analysis were taken every 20 m along the east-west direction and every 16 m along the north-south direction.

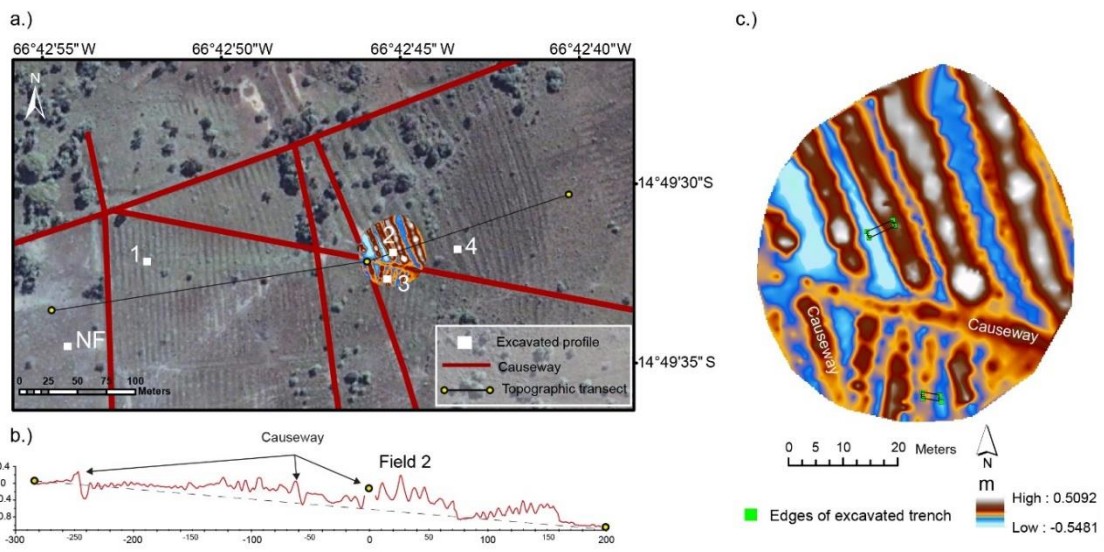

**Figure 7 a.)** Study area including causeways and excavation sites. **b.)** Topographic transect going from west to east. **c.)** Digital elevation model present-day morphology covering Field 2 and 3.





**Figure 8 Down-profile variations of selected elements and grain size.**





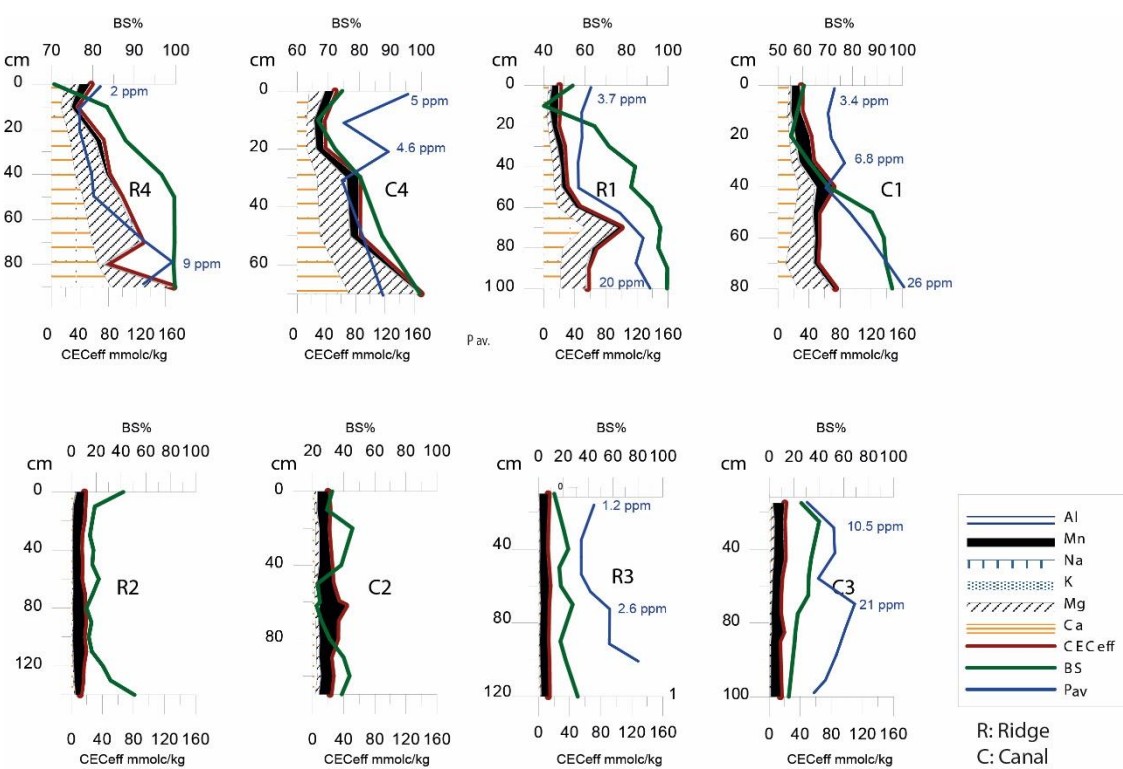

**Figure 9 Down-profile variations of available Cations and Phosphorous.**

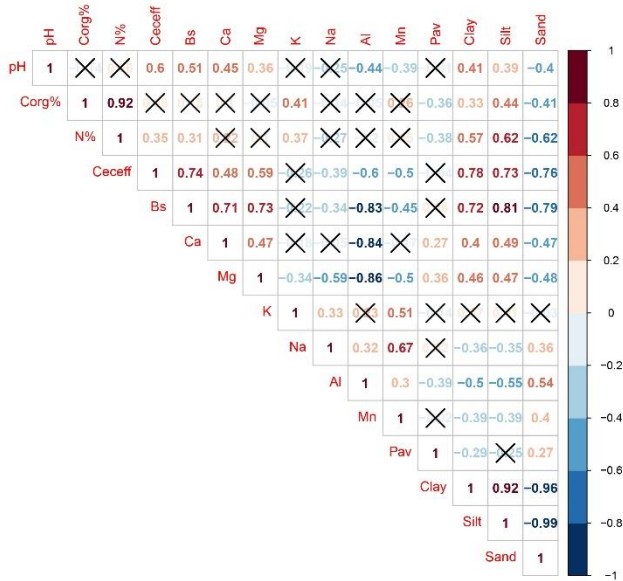

**Figure 10 Pearson Correlation matrix. Values with significance level p > 0.05 are crossed out.**





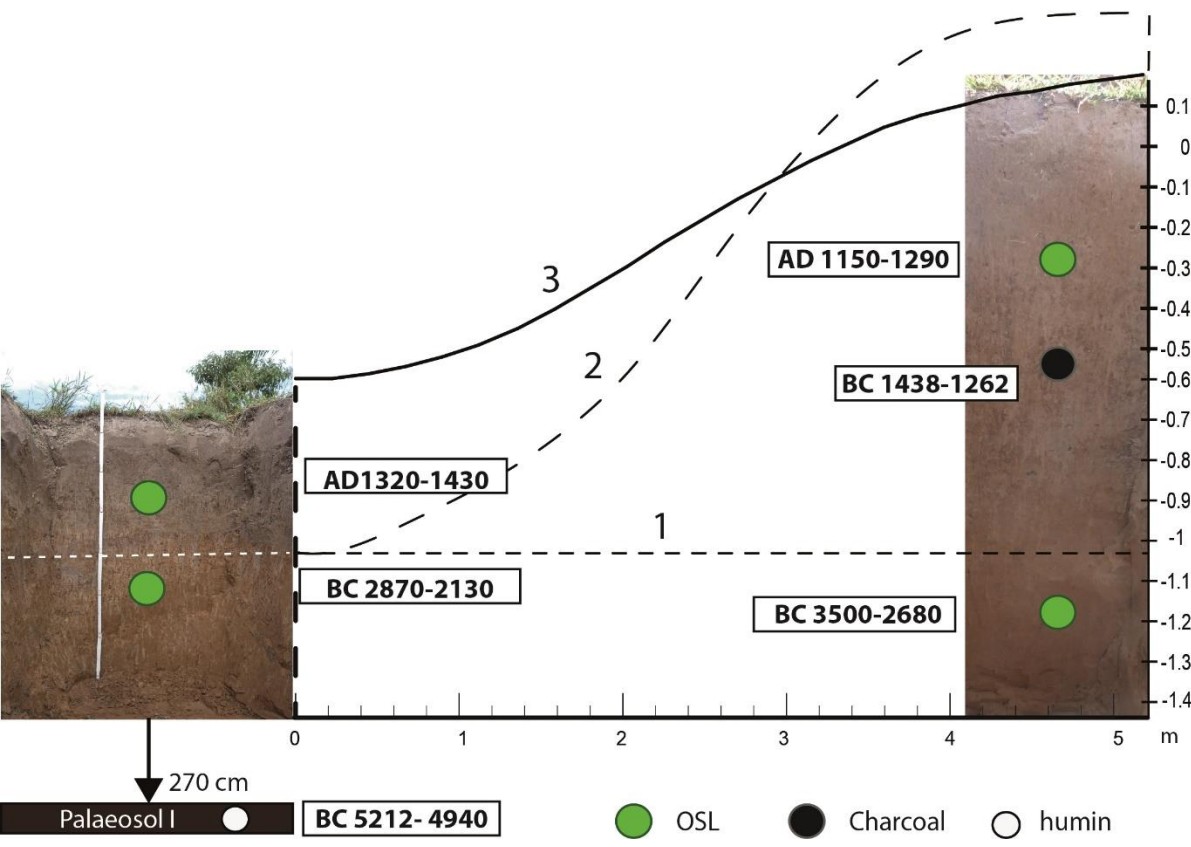

**Figure 11 Reconstruction of Field 2 illustrating the three suggested different phases: 1 ancient surface before the construction of the fields (short dashed line), 2: Original field profile (long dashed line) and 3: present day profile of abandoned field (solid line).**

