# Peer review of "An insight into pre-Columbian raised fields: The case of San Borja, Bolivian lowlands"

_SOIL, 2016_

## Referee Comment (RC1) · Anonymous Referee #1 · 6 Jun 2016

This manuscript deals with the question of raised field in the pre-columbian period. Authors have studied raised fields of different sizes built near the town of San Borja to provide new insights into their morphology, functionning and time frame of their use. For that purpose, they have used a multidisciplinary approach combining topographic surveying and mapping, soil analyses and dating (OSL and radiocarbon) of the raised fields sediments. Authors aim to answer three questions concerning environmental context (morphology of raised fields), possible link between dimension and soil properties of raised fields, and establishment of a chronology of raised field use.

This paper is written clearly, well-documented (11 figures are provided) and the methodology seems rigourous. I'm not a specialist of soil geochemistry and can't comment in detail about this part of the paper. About dating, the methodology is good, the main data are provided and the results are well-interpreted.

From my point of view, this is a significant paper which answers the questions it raises. Nonetheless, the comments of a geochemist or a sedimentologist would be useful.

---

## Referee Comment (RC2) · C. Prat (Referee) · 13 Jun 2016

To avoid repetition, I will not repeat what the reviewer1 said about the description of this paper. I'm fully agree about the scientific quality of this paper. It is written clearly, well-documented but I think that figures and photos should be improved. Some tecnical details should be corrected as well as some typing mistakes before publication.

I am surprised that authors with all the information that they have, do not give names to their soils according some soil classification. It sould be good if they do it.

In regards to the question of the fertility of the soils, manure during pre-columbian times does not really exist because cattle appears with the Spanish! So, it is more

accurate to think fertility in terms of soil fertility, plant rotations, plant associations, fallox, slash and burn systems... In that way, it should be very usefull if authors could give information about plants used in ths areas by pre-Columbian populations. This could help to understand/resolved the question of fertility.

Another point that author could discuss is the time spend to build a raised field oh 1 ha for instance. Is it possible to do it by few men in few days? weeks? monthes? Answering to that question will give an argument about the land use occupation: if you need a lot of time, energy etc to build this kind of raised field, obviously you will not abandon it few monthes after!

Finally, as informations for the authors, it could be usefull for them to get informations from the mexican "chinampas" systems which is still functionning in our days in the south of Mexico city.

Please also note the supplement to this comment:
http://www.soil-discuss.net/soil-2016-27/soil-2016-27-RC2-supplement.pdf

[Figure]

**Supplement:**

**Corrections of the article**

TEXT

1/ Change the word « palaeo » to « paléo » which is more correct, in all the text.

2/ p4 first line, I dont understand the meanning of « with crevasse occuring every few years » if you are speaking of the river ! I guess that you are speaking of soils, so you should change the writing of your sentence. In other end, instead of « crevasse », you should use « crack » which is more usual for this soil feature description.

3/ Change the medical word « avulsion » by a tecnical one as « lifting » for instance.

4/ p5 line 15 : Add a space between Na+, and Mn2+.

5/ p7 line 14 : change « ; » by « : » after « each year ».

6/ Change the capital of Goethite and Lepidocrite with a lowercase letter.

7/ p7 line 23 : change « ; » by « : » after « major layers».

8/p8 line5, join « Fig » and the number « 4 » in the same line.

9/p8 line17, join « 30-50 » with « cm » in the same line.

10/p8 line20, put « meanwhile » before « the content ».

11/p8 line29, join « 20 » with « cm » in the same line.

12/ p9 line 15, Change the capital Quartz with a lowercase letter.

13/p10 line 5, Take off « base saturation ».

14/p10 line 12, Put a point at the end of the sentence.

15/p10 line 25, Put a space before 270 cm.

16/p11 line 15 same comments than above in point 2 and 3 .

17/p13 line 13 same comments than above in point 2.

18/p13 line 19 change « This seems to be common » by « This is common » , because it is just common !

19/p15 line 7, Replace the « . » after « aluminium » by a space.

REFERENCES

P18, line 8, Replace « Dephine » by « Delphine »

P20, line 3, Replace « hidrauclicas » by « hidraulicas »

P20, line 21, Replace « Nino » by « Niño »

P20, line 44, Replace « Nino » by « Niño »

P21, line 5, Replace « Acuático » by « acuático »

P21, line 18, Replace « off Peru » by « of Peru »

TABLES

Replace the #DIV/0 ! which are appearing in some lines of the Ca/Al column by « -«

FIGURES

Fig 1. Legend should explain the green color (= raised fields)

Fig 3. The settlements (close to the pound) presents in Fig 1 is missing here !

Fig 4 : I do not understand the secand image of Ridge2 « yellow oxidized … »

Fig 8 : You must used the same deep scale for all the profile as well as the same X scale.

The meanning of the dot line is not clear at all. They do not correspond to the soil description, for instance ! The boundary of infilling is +- 60 cm according to the profile description and here it is at 35 cm deep !

Fig. 9 : Use the same scale for the deepness of soil profile

---

## Author Comment (AC1) · 28 Jun 2016

We would like to thank you very much for your interest in the revision of our manuscript and for your comments.
* * *

---

## Author Comment (AC2) · 28 Jun 2016

Referee #2 Dear Christian Prat, we highly appreciated your comments and constructive advices. We agree with almost all of them and have revised the paper accordingly. Please find below our response to each of your comments, which will be included in the next version of our manuscript.

1/ Change the word " palaeo " to " paléo " which is more correct, in all the text. Response: To our knowledge palaeo is more common in international literature.

2/ p4 first line, I dont understand the meanning of " with crevasse occuring every few years " if you are speaking of the river ! I guess that you are speaking of soils, so you

should change the writing of your sentence. In other end, instead of " crevasse ", you should use " crack " which is more usual for this soil feature description.

3/ Change the medical word " avulsion " by a tecnical one as " lifting " for instance

Response: The use of the words crevasse and avulsion is correct in this context. Crevasse and avulsion are common words used in the field of geomorphology and description of fluvial landforms. However to make it clearer for non-geomorphologist we added a short definition of each:

"The Río Maniqui is one of the most dynamic rivers in the LM with crevasse occurring every few years, leading to complete river avulsions on a sub decadal time frame (a crevasse describes the process when a river breaks though its river levee ; an avulsion is a natural change of river course that leads to theabandonment of the old channel and the establishment of a new one (Charlton, 2007))."

Reference (Charlton, 2007 ) has been added.

4/ p5 line 15 : Add a space between Na+, and Mn2+

Response: Has been added.

5/ p7 line 14 : change " ; " by " : " after " each year ".

Response: Has been changed.

6/ Change the capital of Goethite and Lepidocrite with a lowercase letter.

Response: Has been changed.

7/ p7 line 23 : change " ; " by " : " after " major layers".

Response: Has been changed.

8/p8 line5, join " Fig " and the number " 4 " in the same line.

Response: Has been changed.

9/p8 line17, join " 30-50 " with " cm " in the same line.

Response: Has been changed.

10/p8 line20, put " meanwhile " before " the content ".

Response: Has been included.

11/p8 line29, join " 20 " with " cm " in the same line.

Response: Has been changed.

12/ p9 line 15, Change the capital Quartz with a lowercase letter.

Response: Has been changed.

13/p10 line 5, Take off " base saturation ". Response: Has been changed.

14/p10 line 12, Put a point at the end of the sentence.

Response: Has been included.

15/p10 line 25, Put a space before 270 cm.

Response: Has been included.

16/p11 line 15 same comments than above in point 2 and 3 .

Response: See point 2 and 3.

17/p13 line 13 same comments than above in point 2.

Response: See point 2 and 3.

18/p13 line 19 change " This seems to be common " by " This is common " , because it is just common!

Response: Has been changed.

19/p15 line 7, Replace the " . " after " aluminium " by a space

Response: Has been replace.

REFERENCES

P18, line 8, Replace " Dephine " by " Delphine " P20, line 3, Replace " hidrauclicas " by " hidraulicas " P20, line 21, Replace " Nino " by " Niño " P20, line 44, Replace " Nino " by " Niño " P21, line 5, Replace " Acuático " by " acuático " P21, line 18, Replace " off Peru " by " of Peru "

Response: Have all be corrected.

TABLES

Replace the #DIV/0 ! which are appearing in some lines of the Ca/Al column by " -" Response: Has been replaced. FIGURES Fig 1. Legend should explain the green color (= raised fields) Response: Has been added. Fig 3. The settlements (close to the pound) presents in Fig 1 is missing here ! Response: Has been changed. Fig 4 : I do not understand the secand image of Ridge2 " yellow oxidized . . . "

Response: This line is discussed in the text. To make it clearer we added (Fig. 4, Ridge 2) in the text.

Fig 8 : You must used the same deep scale for all the profile as well as the same X scale. The meanning of the dot line is not clear at all. They do not correspond to the soil description, for instance ! The boundary of infilling is +- 60 cm according to the profile description and here it is at 35 cm deep !

Response:

1.We changed the Y-axes to the same scale. 2.We would prefer to keep the X-axes we kept as the chosen scale do best show the down profile differences discussed in the text. The values are all provided in the supplementary material (Table S3). 3.We added a description of the dotted line to the legend. The dotted line is the level of ferruginisation (relative accumulation of Iron) discussed in the text.

Fig. 9 : Use the same scale for the deepness of soil profile

Response:

To improve the readability of the figure, we prefer to keep the original scale in Fig. 9. Figure caption and order of the profiles have been changed for improved understanding. Figure 9 Down-profile variations of available cations (CEC), base saturation (BS) and Phosphorous (Pav).

[Figure]

**Fig. 1.** Figure 1: Study site in the Bolivian Lowlands showing raised fields and causeways, including the location of the excavated profiles and locations sampled for grain size analysis

[Figure]

**Fig. 2.** Figure 3 Study area including anthropogenic earthworks, natural geomorphological features and the two farms Campo España and El Progresso.

[Figure]

**Fig. 3.** Figure 8 Down-profile variations of selected elements and grain size.

[Figure]

**Fig. 4.** Figure 9 Down-profile variations of available cations (CEC), base saturation (BS) and Phosphorous (Pav).